# Consistent Symmetry Representation over Latent Factors of Variation

## Abstract

Recent symmetry-based methods on variational autoencoders have advanced disentanglement learning and combinatorial generalization, yet the appropriate symmetry representation for both tasks is under-clarified. We identify that existing methods struggle with maintaining the *consistent symmetries* when representing identical changes of latent factors of variation, and they cause issues in achieving equivariance. We theoretically prove the limitations of three frequently used group settings: matrix multiplication with General Linear groups, defining group action with set of vectors and vector addition, and cyclic groups modeled through surjective functions. To overcome these issues, we introduce a novel method of *conformal mapping* of latent vectors into a complex number space, ensuring consistent symmetries and cyclic semantics. Through empirical validation with ground truth of factors variation for transparent analysis, this study fills two significant gaps in the literature: 1) the inductive bias to enhance disentanglement learning and combinatorial generalization simultaneously, and 2) well-represented symmetries ensure significantly high disentanglement performance without a trade-off in reconstruction error, compared to current unsupervised methods. Additionally, we introduce less guidance-dependent validation results, extending our findings to more practical use. Our research highlights the significant impact of verifying consistent symmetry and suggests required future research for advancing combinatorial generalization and disentanglement learning.

## 1 Introduction

Combinatorial Generalization (Montero et al., 2021) and disentanglement learning (Bengio et al., 2013) have been studied for a common objective of constructing inductive bias for latent factors of variation, mainly discussed on variational autoencoders (VAEs) in Montero et al. (2022; 2021); Schott et al. (2022) and Kingma & Welling (2013); Chen et al. (2018); Kim & Mnih (2018); Keller & Welling (2021a); Jeong & Song (2019); Keller & Welling (2021b); Shao et al. (2020; 2022), respectively with dimension-wise disentangled representation (Wang et al., 2023). The importance of incorporating symmetries in latent vectors, grounded in group theory, has been highlighted in Higgins et al. (2018; 2022); Huh et al. (2023). Moreover, the development of symmetry-based methods has significantly enhanced performance, as demonstrated in Zhu et al. (2021); Yang et al. (2021); Keurti et al. (2023); Tonnaer et al. (2022b); Jung et al. (2024); Hwang et al. (2023).

These studies have primarily focused on incorporating diverse symmetries into models. For example, in disentanglement learning, the Lie group-based works (Zhu et al., 2021; Jung et al., 2024; Keurti et al., 2023) represent the group as a matrix exponential that acts on the set of latent vectors with matrix multiplication. Another approach implements symmetries with the pre-defined groups such as $SO(n)$, $SE(n)$, and $O(n)$ (Winter et al., 2022). Other branches based on cyclic group representations employ a mapping function that projects real number vectors onto a unit circle using a surjective function (Yang et al., 2021; Tonnaer et al., 2022b; Cha & Thiyagalingam, 2023). In combinatorial generalization, MAGANet (Hwang et al., 2023) derives symmetry representations through vector subtraction. However, these works often overlook the critical task of verifying their equivariance, which is essential for capturing all symmetries in latent space. In particular, the equivariance, which manifests as consistent symmetries, is insufficiently induced for identical change of latent factors of the variation that implies the same semantic change, as empirically demonstrated in Hwang et al. (2023).

In this regard, we address limitations in representing the consistent symmetries for the same variations. First, we identify and prove the limitations caused by three conditions of group settings: 1) the use of the General Linear group with matrix exponential for matrix multiplication, 2) vector addition is used to define a group action for cyclic semantics, and 3) surjective mapping from latent vector to unit circle. Subsequently, we propose *conformal mapping* of latent vectors to discretized and cyclic representations that guarantee consistent transformations for all pairs of adjacent points. To guarantee consistency among the mapped representations, we arrange them on a shared, fixed grid (codebook), selecting their nearest codes and performing translations from the canonical point. In the final step, the two points on the grid corresponding to two samples are adjusted to maintain symmetry for the given latent factors of variation in the data.

Through empirical analysis of the consistent symmetries for the ground truths of latent factors of variation in the given data, we aim to answer two important questions that have not yet been clarified in the literature. Firstly, this study queries whether the inductive bias associated with combinatorial generalization and disentanglement learning enhances each performance simultaneously. Secondly, it examines the efficacy of inductive biases achieved through refined symmetry expressions in disentanglement learning. These studies highlight the impact of correctly expressing symmetries as an inductive bias, suggesting a necessary direction for this field. In addition, we also explore the potential of these approaches under more practical conditions, reducing the reliance on ground truth.

Our main contributions are:

1. identifying the difficulty of representing consistent symmetries for the identical change of latent factors of variation in current disentanglement learning and combinatorial generalization,

2. proposing an expression method using *conformal mapping* to a space that guarantees consistency of general symmetries and even for cyclic semantics,

3. providing a learning method for inducing equivariance while maintaining the consistent symmetries,

4. empirically validating that disentangled representations and combinatorial generalization are improved simultaneously, and investigating achievable disentanglement learning performance without a trade-off of reconstruction error through learning appropriate symmetry representation.

## 2 PROBLEMS OF GROUP SETTINGS IN UTILIZING CONSISTENT SYMMETRIES

### 2.1 WHAT IS THE CONSISTENT SYMMETRY?

**Cyclic Semantics of Dataset Space** We define the group $G_F = G_{F_1} \times G_{F_2} \times \cdots \times G_{F_k}$ and the latent factor $F = F_1 \times F_2 \times \cdots \times F_k$ as a $G_F$-set. Motivated by Yang et al. (2021); Tonnaer et al. (2022a), we assume that each $G_{F_i}$ is a cyclic group $G_{F_i} = \{e, g_{F_i}, g_{F_i}^2, \ldots, g_{F_i}^n\}$ as shown in Fig. 1 *e.g.*, $g_{F_i} \in$ {change the shape, move right 1 step, move up 1 step, etc.}. Also, we assume that there is an isomorphism between latent factor and dataset space $\Omega : \mathcal{F} \to \mathcal{X}$. We then define that dataset $X$ consists of *cyclic semantics* when dataset $X$ is generated from the cyclic latent factors ($F$) through

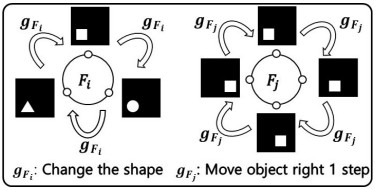

Figure 1: Example of cyclic semantics of the dataset.

isomorphism $\Omega$. We define the *identical change of latent factors of variation* ($g_{F_i}$), represented by a group element, as the transformations between samples are the same as shown in Fig. 2 (middle side).

**Disentangled Representation on Latent Vector Space** As defined, the disentangled representations with group theory (Higgins et al., 2018), we follow the definition, where equivariant function $q_\phi$ is defined as $q_\phi : \mathcal{X} \to \mathcal{Z}$, group $G_z = G_{z_1} \times G_{z_2} \times \cdots \times G_{z_n}$, $G_z$-set as a set of latent vectors $z = z_1 \times z_2 \times \cdots \times z_n$, $G_{z_j} = \{e, g_{z_j}, g_{z_j}^2, \ldots, g_{z_j}^n\}$, and $\mathcal{Z}$ is a latent vector space. Differently, we consider that $G_{z_i}$ only affects a single dimension of latent vector $z_i$ for dimension-wise disentangled representation Wang et al. (2023).

Figure 2: Overview of our process to guide the motivation and expectation. The left side shows inconsistent symmetries, and the right side represents the ideal case of consistent symmetries. As shown in Hwang et al. (2023), the identical change of latent factors of variation $g_{F_j}$ is represented in diverse representations $(g_{z_j}^{(1)\to(2)}, g_{z_j}^{(2)\to(3)}, \ldots, g_{z_j}^{(i)\to(i+1)})$ when latent factors are mapped into latent vector space as shown in left side, where $g_{z_j}^{(i)\to(i+1)}$ is a symmetry between $z_j^i$ and $z_j^{i+1}$.

**Consistent and Inconsistent Symmetry**

**Definition 2.1.** As we define the factor set $F_j = \{F_j^0, F_j^1, \ldots, F_j^k\}$ and group $G_{F_j} = \{e, g_{F_j}, g_{F_j}^2, \ldots, g_{F_j}^k\}$ act on $F_j$ such that $F_j^{i+1} = g_{F_j} \circ F_j^i$. The factor space $\mathcal{F}$, which includes $F_j$, is mapped to a latent space $\mathcal{Z}$ via a composite function $q_\phi \circ \Omega : \mathcal{F} \to \mathcal{Z}$, as used in neural networks. The group action on the factor space $q_\phi \circ \Omega(F_j^{i+1}) = q_\phi \circ \Omega(g_{F_j} \circ F_j^i)$ is translated to $z_j^{i+1} = g_{z_j}^{(i)\to(i+1)} \circ z_j^i$ on the latent space. We define $g_{z_j}^{(i)}$ as a *consistent symmetry* if $g_{z_j}^{(i)\to(i+1)}$ remains identical for all $i$, and a *inconsistent symmetry* otherwise.

## 2.2 DIFFICULTY OF CONSISTENT SYMMETRY FOR DISENTANGLED REPRESENTATION

The purpose of disentanglement learning with symmetries $(g_z \in G_z)$ is to change a single dimension value of latent vectors as symmetries act on the latent vectors. However, there is a lack of sufficient theoretical discourse on which symmetries are suitable for disentangled representations. In this section, we show the difficulty of previous works for consistent symmetry such that 1) only the identity element represents consistent symmetries for cyclic semantics of the dataset with disentangled representations (cases 1 and 2), and 2) symmetry information is not preserved (case 3).

**Case 1: A Limitation of** $GL(n)$  General Linear group $GL(n)$, used in prior works (Kuzina et al., 2022; Miyato et al., 2022; Marchetti et al., 2023a), is limited in representing the consistent symmetries for disentangled representation. We first show the property of disentangled representation with matrix exponential. Then we show the limitation of $GL(n)$.

**Proposition 2.2.** *Let the symmetry group $G_z$ ($GL'(n)$) is defined as a subgroup of the General Linear group that implemented with matrix exponential, where $GL'(n) = \{e^M | M \in \mathbb{R}^{n \times n}\}$, $g^k$ is an element of $GL'(n)$, and $g = \prod_k g^k$. Then $e^g z = e\mathbf{I} g z + v'$.*

**Theorem 2.3.** *(Limit of $GL'(n)$) According to Proposition 2.2, only the identity matrix ($g = \mathbf{I}$) represents the cyclic semantics of the dataset with consistent symmetries, where $g \in GL'(n)$.*

**Theorem 2.4.** *(Limit of $GL(n)$) If $H \subset GL(n)$, then only the identity matrix of $GL(n)$ represents the cyclic semantics of the dataset with consistent symmetries, where $H = \{h | h = \mathbf{I} + M^k\}$, $m^k$ is a column vector of $M^k$, $m^j = \vec{0}$ and $j \in \{1, 2, \ldots, n\} \backslash \{k\}$.*

Therefore, the limitation of $GL(n)$ is that only the $\mathbf{I}$ represents the consistent symmetry with disentangled representation according to the Theorem 2.3, and 2.4. It implies that if $g \neq \mathbf{I}$, then $g$ can not represent the consistent symmetry. More details are in Appendix A.1.

**Case 2: A Limitation of Using Vector Addition**  Another setting that causes problems in maintaining consistent symmetries for cyclic semantics using vector addition is utilized to define a group action between two latent vectors for an equivariant model, as used in Balabin et al. (2024)

**Corollary 2.5.** *If the group action is defined as $\alpha(g, z_i) = g + z_i$, then only zero vector represent the consistent symmetry for cyclic semantics with disentangled representation, where $z \in \mathbb{R}^n$.*

According to the Theorem 2.5, it also shows a limitation in that only the identity element $\vec{0}$ represents the consistent symmetry. More details of the proof are in Appendix A.2.

**Case 3: A Limitation of Surjective Function** The last setting is a surjective function that maps latent vectors to the unit circle (Yang et al., 2021; Tonnaer et al., 2022b; Cha & Thiyagalingam, 2023), causing undifferentiated symmetries under more general latent factors of variation and losing part of the dataset's symmetry information.

**Corollary 2.6.** *By the equivariant and surjective function $b : \mathcal{Z} \to \mathcal{Y}$, the capacity of $\mathcal{Z}$ and $\mathcal{Y}$ is $|\mathcal{Z}| \geq |\mathcal{Y}|$ then $\Gamma'$ is an endomorphism because $|G_{\mathcal{Z}}| \geq |G_{\mathcal{Y}}|$. On the other hand, isomorphism identically maps the two spaces ($|G_{\mathcal{Z}}| = |G_{\mathcal{Y}}|$).*

Therefore, if $b$ is surjective and not injective, then there exists at least one case where $\Gamma'(g_i) = \Gamma'(g_j)$. It implies that loss of symmetry structure occurs with a surjective function. More details of the proof are in Appendix A.3.

## 3 CONFORMAL MAPPING FOR CONSISTENT SYMMETRIES (CMCS)

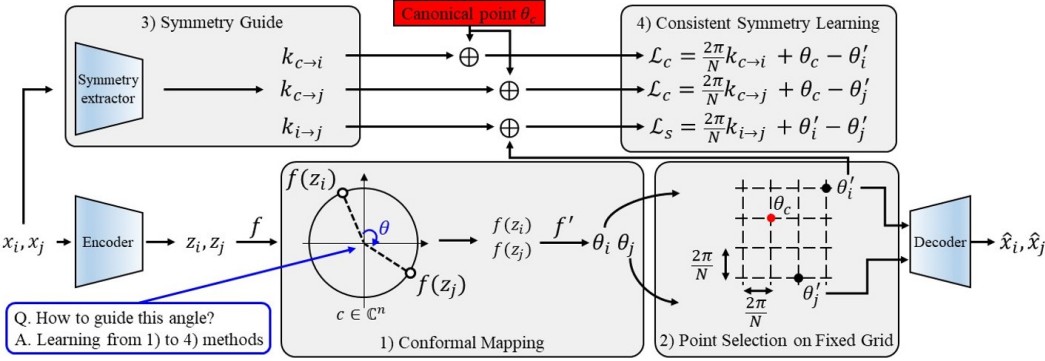

Figure 3: The overall architecture of our proposed method comprises four distinct components: 1) conformal mapping of latent vectors to angle space, 2) point selection of fixed grid for consistent symmetry, 3) defining the step size between two inputs through three methods, and 4) a loss function that satisfies the group action $\alpha(g, \theta) = g + \theta$.

### 3.1 CONFORMAL MAPPING: CYCLIC SEMANTIC AND ISOMORPHISM

We first define the latent vector space as a $G$-set of a cyclic group to ensure the consistent symmetries because a single cyclic group element can represent all elements, as demonstrated in Higgins et al. (2018). To address the issues discussed in Section 2, we implement one of the conformal mappings (Kreyszig et al., 2011) that maps real numbers to complex numbers.

**Cyclic Group to Represent Cyclic Semantics** As we assume that real-world states serve as latent factors of variation for input samples in Section 2.1, the symmetry group $G_F = \mathbb{Z}/|F_1|\mathbb{Z} \times \mathbb{Z}/|F_2|\mathbb{Z} \times \ldots \times \mathbb{Z}/|F_n|\mathbb{Z}$, where $|F_k| \in \mathbb{Z}^+$ represents the number of factors in the datasets. Additionally, the cyclic group effectively represents the symmetries over the same group action, as the cyclic group $G = \{e, g^1, g^2, \ldots, g^{n-1}\}$ consists entirely of integer powers of the group element $g$. Therefore, if the model learns a single symmetry element $g$, it represents the entire symmetry group. Motivated by Yang et al. (2021), we implement the cyclic group as the $n^{th}$ root of unity.

**Group Action and $G^c$-set** As we define the cyclic group as $n^{th}$ root of unity, the cyclic group $G^c = G_1^c \times G_2^c \times \cdots \times G_k^c$ and $G_i^c = \{g_i^c | g_i^c = \frac{2\pi}{N}k, k \in \{0, 1, 2, \ldots, N-1\}\}$, where $N \in \mathbb{Z}^+$. We define the group action $\alpha : \Theta^n \times G^c \to \Theta^n$ as $\alpha(g^c, \theta) = g^c + \theta$, where the latent vector $\theta \in \Theta^n$, and $\Theta = \{\theta | -\pi < \theta \leq \pi\}$.

**Conformal Mapping for Complex Number Space** VAE frameworks establish the latent vector space in the real number space with the Gaussian normal distribution, so the latent vector space is not a $G^c$-set as we assume ($z \in \Theta^n$). For the defined $G^c$-set, we utilize the conformal mapping and

invertible function $f' \circ f : [-\infty, \infty] \rightarrow \{\theta | -\pi < \theta \leq \pi\}$. To map real numbers to the complex number space, we utilize the conformal mapping function (bijective) $f : [-\infty, \infty] \rightarrow \{c_i^k \in \mathbb{C} : |c_i^k| = 1\}$, defined as follows:

$$f(z_i^k) = c_i^k = \frac{z_i^k - i}{z_i^k + i} = i\frac{-2z_i^k}{(z_i^k)^2 + 1} + \frac{(z_i^k)^2 - 1}{(z_i^k)^2 + 1}, \tag{1}$$

where the $z_i^k$ is a $k^{th}$ dimension value of $z_i \in \mathbb{R}^n$. We define a bijective function that maps complex numbers to the angle space for simplicity $f' : \{c_i^k \in \mathbb{C} : |c_i^k| = 1\} \rightarrow \{\theta_i^k | -\pi < \theta_i^k \leq \pi\}$ as follows:

$$\theta_i^k = f'(c_i^k) = \begin{cases} \cos^{-1}(\Re(c_i^k)) - \pi, & \text{if } \Im(c_i^k) >= 0 \\ \pi - \cos^{-1}(\Re(c_i^k)) & \text{otherwise} \end{cases}, \tag{2}$$

where $\Re(c_i^k)$ and $\Im(c_i^k)$ are real and imaginary parts of $c_i^k$, respectively.

## 3.2 Point Selection on Fixed Grid: Fixed Codebook for Consistent Symmetries

The angle space $\Theta$ is a continuous space as defined by the bijective function $f' \circ f$. As we define a finite cyclic group $G^c$ acts on the latent vector space ($G^c$-set), we utilize a fixed grid instead of a learnable grid (Hsu et al., 2023) to fix the unit symmetry $\hat{g}_i^c$ as shown in Fig. 3 fixed grid selection box, where the interval between two nearest codes is always $\frac{2\pi}{N_i}$ ($\hat{g}_i^c = \frac{2\pi}{N_i}$). We utilized the finite scalar quantization (Mentzer et al., 2024) for fixed codebook $V \in \mathbb{R}^N$ as follows:

$$V = [-\pi + \frac{2\pi}{N}, \cdots, -\pi + \frac{2\pi}{N}k, \cdots, -\pi + \frac{2\pi}{N}(N - 1), \pi]. \tag{3}$$

Then we select the nearest neighbor of the latent vector as Hsu et al. (2023): $\theta'^k = \arg\min_{V^i \in V} |V^i - \theta^k|$, where $V^i$ is the $i^{th}$ dimension value of the codebook $V$. We define the grid loss $\mathcal{L}_{grid} = ||\theta' - \theta||_2^2$ to consistently select the gird, where $|| \cdot ||_2$ is a L2 norm.

## 3.3 Symmetry Guide: How to Guide the Step ($k$)?

As we define the cyclic group $G_i^c = \{g_i^c | g_i^c = \frac{2\pi}{N_i}k, k \in \{0, 1, 2, \ldots, N_i - 1\}\}$, we implement the step $k$ to guide how much step moves to be $\theta_i = g^c + \theta_j$ (defined group action). We propose three guiding approaches: 1) ground truth, 2) supervised, 3) and semi-supervised methods.

**Ground Truth Based Method**   As shown in Fig. 3, we set the symmetry group elements $g^c$ from the ground truth of samples as follows:

$$k_{i \rightarrow j} = \begin{cases} l_j - l_i & \text{if } l_j - l_i \geq 0 \\ N + l_j - l_i & \text{otherwise} \end{cases}, \tag{4}$$

where $k_{i \rightarrow j}$ is a step size, $g^c = \frac{2\pi}{N}k_{i \rightarrow j}$, and $l_i$ and $l_j$ are labels of samples $x_i$ and $x_j$, respectively.

**Supervised Method**   As shown in Fig. 3, we train the symmetry extractor to predict the labels of samples ($\hat{l}$) using cross-entropy loss, defined as $\mathcal{L}_{pred} = C.E.(\hat{l}, l)$. We then define the symmetry group elements by $\hat{l}$ instead of the ground truth labels $l$.

**Semi-Supervised Method**   We utilize a $p$ ratio of the labels for prediction, while the remaining labels are predicted using the pseudo-label loss as follows: $\mathcal{L}_{pl} = \sum_i^{|F_i|} D_{\text{KL}}(p(l^i|x)||p(l^i))$, where the $l^i$ is a $i^{th}$ factor of the label and a discrete uniform distribution $p(l^i) \sim \mathcal{U}\{1, |F_i|\}$. We define the $p(l^i|x)$ as the distribution of the classifier.

## 3.4 Consistent Symmetry Learning: Objective Function

**Symmetry Loss (Relative Position)**   As we define the group action $\alpha(g^c, \theta_i)$ and step size $k$, we implement the symmetry loss $\mathcal{L}_s$ to satisfy $\theta_j = \frac{2\pi}{N}k_{i \rightarrow j} + \theta_i$:

$$\mathcal{L}_s = ||f' \circ f \circ q_\phi(x_j) - (\frac{2\pi}{N}k_{i \rightarrow j} + f' \circ f \circ q_\phi(x_i))||_2^2. \tag{5}$$

We set the code loss $\mathcal{L}_{code} = \mathcal{L}_{grid} + \mathcal{L}_s$.

**Canonical Loss (Absolute Position)** The defined symmetry $g^c$ represents the movement between two latent vectors ($\boldsymbol{\theta}_i$ and $\boldsymbol{\theta}_j$). It implies that learning symmetries depends on pairs of observations. To eliminate this dependency on specific observations, we propose learning absolute transformations through a learnable canonical point $\boldsymbol{\theta}_c$ to satisfy $\boldsymbol{\theta}_i = \frac{2\pi}{N}\boldsymbol{k}_{c\rightarrow i} + \boldsymbol{\theta}_c$:

$$\mathcal{L}_c = ||f' \circ f \circ q_\phi(x_i) - (\frac{2\pi}{N}\boldsymbol{k}_{c\rightarrow i} + \boldsymbol{\theta}_c)||_1, \tag{6}$$

where $\boldsymbol{\theta}_c$ is a learnable canonical point $\boldsymbol{\theta}_c \in \boldsymbol{\Theta}^n$, $\boldsymbol{k}_{c\rightarrow i} = l_i$, and $|| \cdot ||_1$ is a L1 norm.

**Objective Loss** Our objective losses are defined as 1) $\mathcal{L}_{GT} = \mathcal{L}_{recont} + \alpha\mathcal{L}_{code} + \gamma\mathcal{L}_c$ for Ground Truth model (CMCS-GT), 2) $\mathcal{L}_{Sup} = \mathcal{L}_{GT} + \beta\mathcal{L}_{pred}$ for supervised method (CMCS-SP, and 3) $\mathcal{L}_{Semi-Sup} = \mathcal{L}_{Sup} + \lambda\mathcal{L}_{pl}$ for semi-supervised method (CMCS-semi).

Table 1: Combinatorial generalization performance of dSprites, 3D Shapes, and MPI3D datasets. We select the best results from the hyper-parameter tunings. The evaluation metric is the reconstruction error (BCE loss for dSprites, and MSE loss for 3D Shapes and MPI3D).

| Method | dSprites | | 3D Shapes | | MPI3D | |
|---|---|---|---|---|---|---|
| | R2E | R2R | R2E | R2R | R2E | R2R |
| $\beta$-VAE | 10.85($\pm$0.67) | 179.52($\pm$12.15) | 16.59($\pm$1.72) | 268.59($\pm$76.59) | 6.63($\pm$0.65) | 8.50($\pm$0.55) |
| $\beta$-TCVAE | 10.73($\pm$0.03) | 153.75($\pm$7.65) | 14.74($\pm$0.14) | 221.72($\pm$41.57) | 5.60($\pm$0.21) | 8.73($\pm$0.36) |
| VAE-MAGA | 11.22($\pm$0.48) | 178.39($\pm$11.64) | 18.84($\pm$3.32) | 213.26($\pm$41.76) | 5.43($\pm$0.59) | 8.44($\pm$0.44) |
| CMCS-SP | **7.24**($\pm$0.94) | **135.70**($\pm$16.48) | **10.23**($\pm$0.67) | **108.44**($\pm$5.82) | **2.92**($\pm$0.03) | **4.16**($\pm$0.14) |
| CMCS-GT | **8.56**($\pm$0.40) | **123.02**($\pm$10.84) | **9.29**($\pm$0.34) | **114.89**($\pm$11.80) | **2.56**($\pm$0.19) | **5.26**($\pm$0.25) |

## 4 EXPERIMENTS

**Common Datsets** We utilize the dSprites (Matthey et al., 2017), 3D Shapes (Burgess & Kim, 2018), and MPI3D (Gondal et al., 2019) datasets for combinatorial generalization and disentanglement learning tasks. More details are in Appendix B.2.

### 4.1 COMBINATORIAL GENERALIZATION

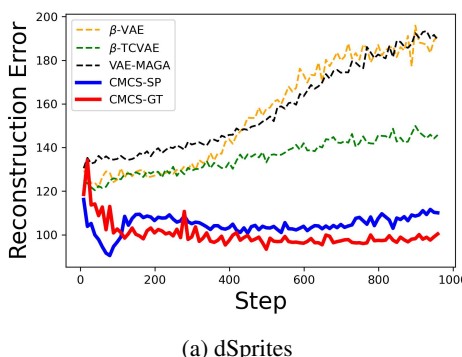

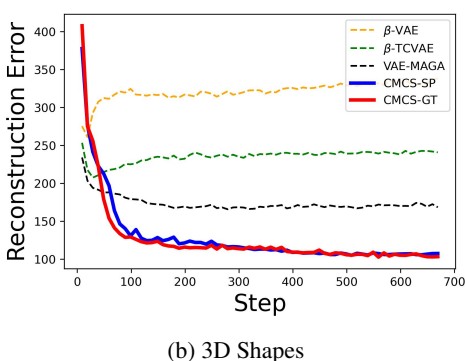

(a) dSprites

(b) 3D Shapes

Figure 4: Reconstruction error on the test set during training.

**Settings** We excess Recombination-to-Element (R2E) and the Recombination-to-Range (R2R) tasks. We separate the training and test datasets following previous studies (Montero et al., 2021; Hwang et al., 2023), with additional details and hyper-parameter tuning provided in Appendix B.3. We run each model with three seeds $\in \{1, 2, 3\}$. We assess the reconstruction error, a general combinatorial generalization metric.

**Quantitative Analysis** As shown in Table 1, the proposed method CMCS-GT is significantly improved with all datasets. It implies the given consistent symmetry from the labels impacts to combinatorial generalization. Also, the supervised method CMCS-SP demonstrates advancements in all datasets, as shown in Table 1.

**Stability of Generalization** As demonstrated in Fig. 4, the baselines do not positively affect generalization during training on dSprites and 3D shapes datasets; in most cases, performance

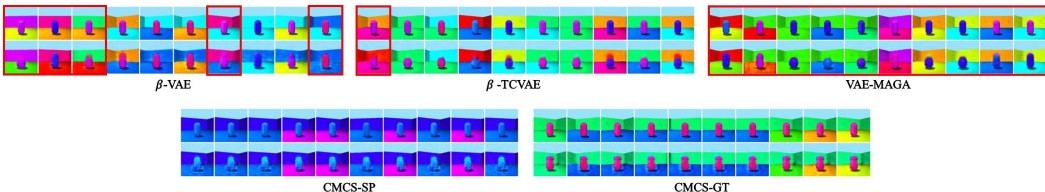

(a) Visualization of generated images of the worst 10 samples of the R2R task. Each $1^{st}$ and $2^{nd}$ row of models shows the group truth samples and the generated results, respectively. The red box indicates the negative results, which do not contain all the semantics of the ground truth. We utilize randomly selected pivot images as introduced in Hwang et al. (2023) for the CMCS model.

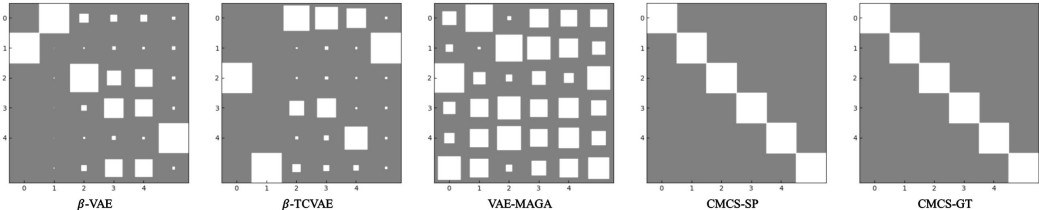

(b) Visualization of DCI metric of the 3D Shapes dataset (training set). A more sparse matrix implies a better disentangled representation. The x-axis refers to the index of the latent vector. The y-axis represents the factor of the dataset, from 1 to 6, corresponding to floor hue, wall hue, object hue, scale, shape, and orientation.

Figure 5: Qualitative results of combinatorial generalization of 3D Shapes dataset.

deteriorates. Conversely, our model positively impacts generalization throughout the training process. It shows 1) the necessity of inductive bias because injecting inductive bias methods (CMCS-SP, CMCS-GT, and VAE-MAGA) improve the generalization as shown in Table 1, and 2) the necessity of consistent symmetries because our method gradually enhances the generalization performance as shown in Fig. 4.

**Qualitative Analysis** As illustrated in Fig. 5a, both proposed methods preserve the semantics of the ground truth while baselines struggle to retain the semantics of unseen samples (ground truth). Comparing the VAE-MAGA and our models, forcing the consistent symmetries method has a much greater impact on generalization. Additionally, our models exhibit a disentangled representation compared to the baselines, as shown in Fig.5b. This implies that a disentangled representation incorporating the symmetry structure promotes combinatorial generalization. Details of other results on the dSprites and MPI3D dataset are provided in Appendix C.

**Prediciton of Unseen Samples** As shown in Fig. 6b and 6c, the distance between the unseen vector (red) and the expected vector (blue) of the proposed method is smaller than the baseline. This explains why the CMCS shows a better result of combinatorial generalization, as shown in Fig. 5a. Also, our model shows the benefit of alignment of factors over the axis. By the definition of a disentangled representation, previous and our works expect that each dimension of the latent vector represents a single latent factor of variation. In this perspective, the dashed line in Fig. 6d shows the ideal case of disentangled representation because the dashed lines of the same color are parallel. From this perspective, our model shows better alignment because the dashed lines of the Tails are parallel, and the dashed lines of the Heads are nearly parallel, as shown in Fig. 6f. Conversely, the baseline dashed lines of both the Heads and the Tails are crossed. These results also show that factors consistently aligned with the dimension of the latent vector space improve the generalization as shown in Fig. 4, and 5a.

### 4.2 DISENTANGLEMENT LEARNING

**Settings** We set the common hyper-parameters of the proposed method $\alpha \in \{100, 1000\}$, $\gamma = 1$ for supervised and ground truth model, $\beta \in \{1.0, 2.0\}$ for supervised method, and $\lambda = 1.0, p = 0.5$ for semi-supervised method. We run 10 seed variance over each model with seed $\in \{1, 2, \ldots, 10\}$. More details are in Appendix B.4.

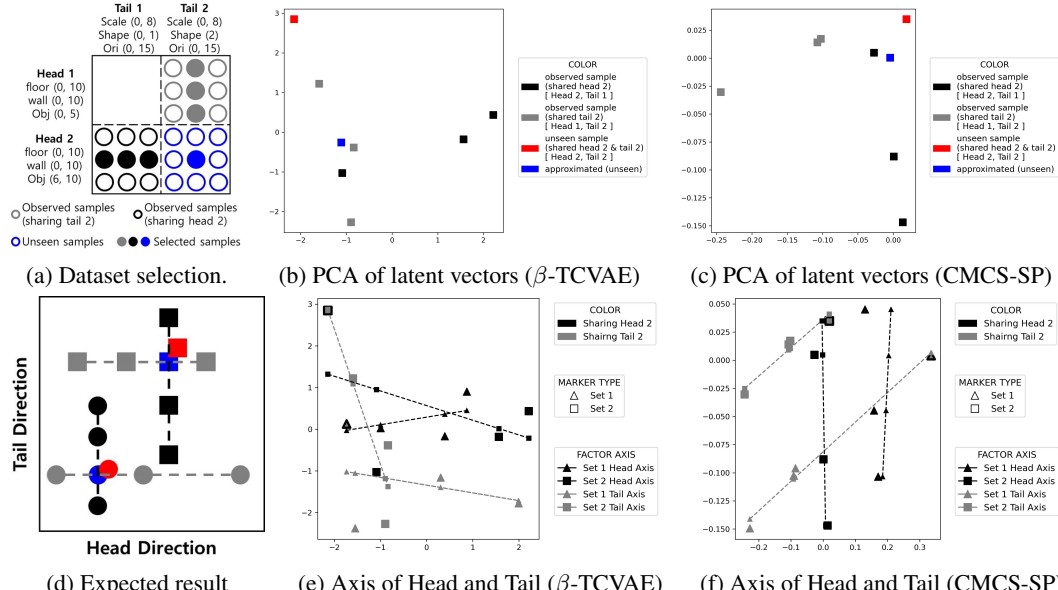

(a) Dataset selection.    (b) PCA of latent vectors ($\beta$-TCVAE)    (c) PCA of latent vectors (CMCS-SP)

(d) Expected result    (e) Axis of Head and Tail ($\beta$-TCVAE)    (f) Axis of Head and Tail (CMCS-SP)

Figure 6: 1) Fig. 6a shows how the dataset is composed and how we select the observed and unseen samples of the 3D Shapes dataset. Head and Tail are sets that consist of a combination of latent factors. 2) Fig. 6d shows the expected result. Each marker, except the blue marker, is a latent vector of selected samples reduced by PCA (principal component analysis). The black, gray, and red markers with the same shape consist of [shared one sample of Head 2, different samples of Tail 1], [different samples of Head 1, shared one sample of Tail 2], and [shared one sample of Head 2, shared one sample of Tail 2], respectively. The blue marker is an expected vector of the unseen sample by the assumption that the model ideally outputs the disentangled representation because the same factors combination is mapped into the same value of vector space as Fig. 6a. 3) Each dashed line is a linear regression of markers sharing the same color and shape. It refers to the alignment of factors over the axis. 4) Fig. 6b, and 6c show the distance between unseen and expected vectors (red and blue). We plot the expected vectors as follows: i) estimate each variance of the axis of Head and Tail samples, ii) assume the smallest variance of the axis as a dominant axis of Head or Tail, *e.g.*, if the variance of the x-axis of Tail samples is the smallest, then x- and y-axis refer to Tail and Head axis, respectively. iii) Then we plot the average of each axis of Head and Tail as an expected latent vector. 5) Fig. 6e, and 6f show the alignment of Head and Tail over the axis.

**Disentanglement Learning Performance** Ground Truth model (CMCS-GT) presents an achievable upper bound in nearly all metrics for the models as shown in Table 2. The supervised method (CMCS-SP)shows results that either approximate or surpass the performance indicated by CMCS-GT, where the DCI score of dSprites and 3D Shapes datasets are higher than CMCS-GT's. The semi-supervised method, using only $50\%$ of the labels, demonstrats performance close to that of the supervised method in some metrics and comparable to that of Ada-GVAE.

Table 2: Disentanglement performance on dSprites, 3D Shapes, and the MPI3D dataset. (Unsup: unsupervised method, Semi-sup: Semi-supervised, Sup: supervised, GT: Ground-Truth, CMCS: our method). Bold text indicates a higher value than the other baseline models.

| type | Method | dSprites | | | | 3D Shapes | | | | MPI3D | | | |
|---|---|---|---|---|---|---|---|---|---|---|---|---|---|
| | | beta-VAE | FVM | MIG | DCI | beta-VAE | FVM | MIG | DCI | beta-VAE | FVM | MIG | DCI |
| Unsup | $\beta$-VAE | 78.40(±9.03) | 64.84(±11.40) | 14.52(±9.33) | 22.37(±11.80) | 90.33(±7.42) | 72.63(±19.55) | 40.49(±23.31) | 54.32(±16.45) | 57.60(±7.93) | 40.86(±3.92) | 4.91(±1.43) | 22.29(±1.42) |
| | $\beta$-TCVAE | 81.80(±11.91) | 70.16(±12.41) | 19.03(±9.40) | 30.89(±8.96) | 88.20(±7.91) | 74.45(±14.68) | 43.17(±28.28) | 59.71(±14.79) | 55.40(±9.52) | 40.80(±2.60) | 5.23(±1.96) | 21.47(±2.35) |
| | Factor-VAE | 87.20(±7.50) | 76.80(±7.50) | 24.98(±12.03) | 33.38(±12.27) | 90.00(±7.87) | 80.85(±13.62) | 62.42(±26.94) | 73.77(±14.08) | 54.00(±7.18) | 39.64(±3.81) | 4.34(±0.69) | 21.24(±2.04) |
| | CLG-VAE | 88.40(±5.80) | 82.21(±5.73) | 20.89(±7.40) | 29.96(±7.05) | 86.20(±5.61) | 77.36(±7.99) | 50.39(±12.37) | 59.25(±11.21) | 46.40(±7.35) | 37.31(±2.27) | 20.77(±5.70) | 24.26(±2.73) |
| Semi-sup | Ada-GVAE | 83.60(±2.61) | 83.67(±2.97) | 21.34(±5.35) | 47.26(±1.89) | 72.75(±6.50) | 59.81(±6.14) | 24.77(±7.48) | 64.57(±4.04) | 64.89(±7.22) | 46.10(±3.19) | 22.48(±8.14) | 41.30(±7.00) |
| | CMCS-semi | 87.00(±7.07) | **84.50**(±1.41) | **31.95**(±2.40) | 39.36(±1.49) | **95.00**(±7.07) | **88.81**(±13.17) | 57.94(±16.52) | 72.14(±3.23) | **67.50**(±12.68) | **81.59**(±3.80) | **61.07**(±4.81) | **78.15**(±0.57) |
| Sup | CMCS-SP | **91.40**(±4.99) | **93.74**(±1.82) | **51.02**(±2.42) | **64.69**(±1.55) | **100.00**(±0.00) | **100.00**(±0.00) | **96.95**(±0.18) | **99.99**(±0.01) | **86.40**(±8.63) | **99.96**(±0.08) | **62.78**(±6.95) | **88.06**(±1.66) |
| GT | CMCS-GT | **95.80**(±4.57) | **99.26**(±1.12) | **51.81**(±2.97) | **63.26**(±2.73) | **100.00**(±0.00) | **100.00**(±0.00) | **96.57**(±0.80) | **99.94**(±0.18) | **76.80**(±9.66) | **99.03**(±0.98) | **65.17**(±8.11) | **81.12**(±1.03) |

**Disentanglement vs. Reconstruction** Most disentangled representation learning models face a trade-off between reconstruction error and disentanglement metrics (Kingma & Welling, 2013; Chen et al., 2018; Higgins et al., 2017; Kim & Mnih, 2018; Zhu et al., 2021; Locatello et al., 2020;

Keurti et al., 2023). However, our model overcomes this trade-off between the two factors, as demonstrated in Fig. 7. Although our model's reconstruction error is two times lower than the baselines, it achieves higher disentanglement performance than the others with MPI3D datasets. Further details are provided in Appendix D.1.

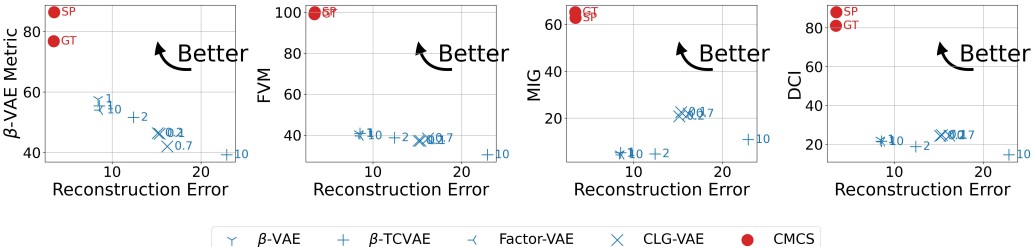

Figure 7: Reconstruction error vs. evaluation metrics of the MPI3D dataset ($\beta$-VAE metric, FVM, MIG, and DCI). The top left side indicates the best results on both objectives

**Case Studies: Impact of Conformal Mapping** We have demonstrated that $GL(n)$, vector addition, and surjective functions are limited in preserving the dataset's symmetry structure. Consequently, we have adopted these three types of symmetry instead of our method. As indicated in Table 3, the CMCS method outperforms other methods across all metrics. This suggests that enforcing the consistent symmetries in the angle space is a more suitable method for disentanglement learning.

Table 3: Performance of the three cases of group settings compared to conformal mapping. $GL(n)$ indicates the General Linear group, Add. indicates a vector addition, and Sur. indicates a surjective method instead of conformal mapping.

| Symmetry | 3D Shapes | | | | dSprites | | | |
|---|---|---|---|---|---|---|---|---|
| | beta-VAE | FVM | MIG | DCI | beta-VAE | FVM | MIG | DCI |
| $GL(n)$ | 73.50($\pm$17.92) | 66.16($\pm$15.95) | 19.66($\pm$24.29) | 37.93($\pm$23.92) | 65.11($\pm$3.89) | 47.69($\pm$8.04) | 2.12($\pm$1.73) | 5.90($\pm$2.11) |
| Add. | 78.60($\pm$11.43) | 60.50($\pm$15.29) | 25.13($\pm$17.12) | 45.84($\pm$8.36) | 68.89($\pm$11.45) | 68.31($\pm$9.48) | 7.22($\pm$4.31) | 11.83($\pm$3.85) |
| Sur. | 73.27($\pm$9.97) | 63.85($\pm$6.81) | 6.20($\pm$4.03) | 29.79($\pm$9.86) | 80.60($\pm$10.83) | 60.40($\pm$6.47) | 13.68($\pm$3.86) | 23.01($\pm$2.28) |
| Ours | **100.00**($\pm$0.00) | **100.00**($\pm$0.00) | **96.57**($\pm$0.80) | **99.94**($\pm$0.18) | **95.80**($\pm$4.57) | **99.26**($\pm$1.12) | **51.81**($\pm$2.97) | **63.26**($\pm$2.73) |

**Qualitative Analysis** As shown in Fig. 8a- 8c, the baseline results show that multiple factors are changed when a single dimension value is changed. On the other hand, our results show the fully disentangled results represent: $1^{st}$ row is floor color changes, $2^{nd}$ row is wall color changes, $3^{rd}$ row is object color changes, $4^{th}$ row is scale of object, $5^{th}$ row is object shape changes, and $6^{th}$ row is the rotation changes.

As shown in Fig. 8d- 8f, the baseline results show that multiple factors are changed when a single dimension value is changed. Also, objects disappear at certain intervals in the baseline results. On the other hand, supervised methods show better results than the baselines: $1^{st}$ row is object color changes, $2^{nd}$ row is object shape changes, $3^{rd}$ row is object size changes, $5^{th}$ row is background color changes, $6^{th}$ row is horizontal axis, $7^{th}$ row is vertical axis changes, and $9^{th}$ row is camera height changes. Also, GT model results represents: $1^{st}$ row is object color changes, $2^{nd}$ row is object shape changes, $3^{rd}$ row is object size changes, $4^{th}$ row is camera height changes, $5^{th}$ row is background color changes, $6^{th}$ row is horizontal axis changes, $8^{th}$ row is object color changes, and $9^{th}$ row is vertical axis changes. Compared to the baseline model, the cases of overlapping factors in a single dimension are reduced by the proposed models.

## 5 RELATED WORKS

**Combinatorial Generalization** Recent research in Combinatorial Generalization shows that models trained on disentanglement learning and verified through ground truth experimentally demonstrate that high disentangled representation does not necessarily imply combinatorial generalization (Montero et al., 2021; Schott et al., 2022; Montero et al., 2022). Differently, we consider the symmetry-based disentangled representations, recently studied in the disentanglement learning field to preserve the symmetry structure of the dataset in latent vector space. MAGANet (Hwang et al., 2023) dramatically improved combinatorial generalization performance by learning symmetries with the Glow

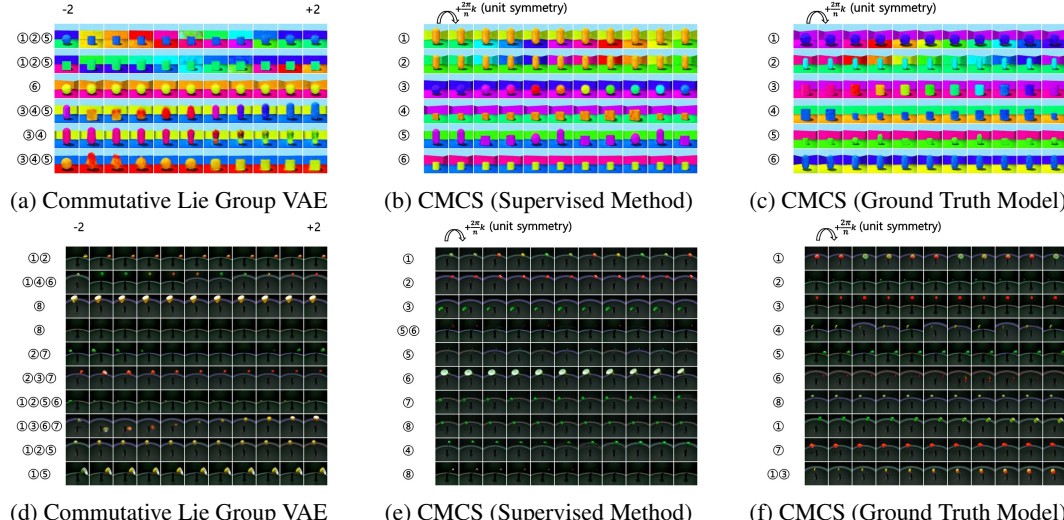

(a) Commutative Lie Group VAE  (b) CMCS (Supervised Method)  (c) CMCS (Ground Truth Model)

(d) Commutative Lie Group VAE  (e) CMCS (Supervised Method)  (f) CMCS (Ground Truth Model)

Figure 8: The $1^{st}$ column images are randomly selected from the dataset. Each row indicates each dimension of each model. $\beta$-VAE, $\beta$-TCVAE, and the Commutative Lie Group VAE trace each dimension value from -2 to +2. The proposed methods apply a group action $+\frac{2\pi}{n}$ to the selected images a total of 10 times. The numbers located on Fig. 8a-8c refer to factors of the 3D Shapes dataset: ①, ②, ③, ④, ⑤, and ⑥ refer to floor hue, wall hue, object color, scale, shape, and rotation, respectively. The numbers in Fig. 8d-8f refer to factors of the MPI3D dataset: ①, ②, ③, ④, ⑤, ⑥, and ⑦ refer to object color, object shape, object size, camera height, background color, horizontal axis, and vertical axis, respectively.

model. Differently, we guarantee a consistent symmetry of identical transformations with CNN based model.

**Disentanglement Learning** The initially proposed methods, such as Higgins et al. (2017); Chen et al. (2018); Kim & Mnih (2018), partition each dimension to ensure mutual exclusivity, employing measures like mutual information or total correlation. However, these approaches do not account for the symmetry structure of the dataset space. Defining the symmetry group and group action as a General Linear group and matrix multiplication (Zhu et al., 2021; Jung et al., 2024; Marchetti et al., 2023b) enhances disentanglement performance. Nevertheless, we theoretically demonstrate the limitations of the General Linear group for cyclic semantics in the disentangled space. Other works commonly define the symmetry group acting on the latent vector space as a cyclic group with surjective functions (Yang et al., 2021; Keurti et al., 2023; Falorsi et al., 2018). Differently, our focus is on employing isomorphism to represent the cyclic group rather than a homomorphism.

## 6 CONCLUSION

In this paper, we address the limitations in expressing consistent symmetry for identical semantic changes, demonstrating three conditions of group settings causing them in existing methods of disentanglement learning and combinatorial generalization with VAEs. We propose a conformal mapping of latent vectors to a space specifically designed to guarantee consistency, along with its learning methodology. This work aims to fill theoretical gaps in the literature by conducting empirical analyses based on given ground truths concerning latent factors of variation. To the best of our knowledge, we are the first to propose a method that enhances combinatorial generalization and disentanglement learning simultaneously. Moreover, the consistent symmetry significantly enhances disentanglement scores for VAEs without compromising reconstruction error, thereby identifying achievable performance for unsupervised methods in case accurate estimation of factor variation is supported. We believe this work indicates a promising direction for constructing more effective inductive biases for disentangled representation in practical generative models.

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

Table 4: Notation Table

| | Set | | |
|---|---|---|---|
| $F$ | Set of latent factor | $X$ | Dataset |
| $Z$ | Set of latent vector | $\mathcal{F}$ | Space of latent factor |
| $\mathcal{X}$ | Space of dataset | $\mathcal{Z}$ | Space of latent vector |
| $\mathcal{Y}$ | Space of latent vector | $\Theta$ | Set of angle: $\{\theta| -\pi < \theta \leq \pi\}$ |
| | **Group** | | |
| $G$ | Group | $G_F$ | Group acted on the set of latent factors |
| $G_z$ | Group acted on the set of latent vectors | $G^c$ | Cyclic group |
| $GL(n)$ | General Linear group | $GL'(n)$ | General Linear group implemented by matrix exponential $GL'(n) \subset GL(n)$ |
| $g$ | Group element of group $G$ | $\mathfrak{g}$ | Lie algebra of $GL(n, \mathbb{R})$ |
| $\alpha(\cdot, \cdot)$ | Group action | $\circ_F$ | Binary operation of Group $G_F$ |
| $\circ_z$ | Binary operation of Group $G_z$ | $g_{z_j}^{(i)\to(i+1)}$ | Symmetry between $\boldsymbol{z}_j^i$ and $\boldsymbol{z}_j^{i+1}$ |
| | **Function** | | |
| $\Omega$ | $\mathcal{F} \to \mathcal{X}$ | $q_\phi$ | $\mathcal{X} \to \mathcal{Z}$ |
| $\Gamma$ | $G_F \to G_z$ | $\Gamma'$ | $G_z \to G_y$ |
| $e^x$ | Matrix exponential | $f$ | $\mathbb{R} \to \{c|c \in \mathbb{C}, |c| = 1\}$ |
| $f'$ | $\{c|c \in \mathbb{C}, |c| = 1\} \to \Theta$ | $C.E.(\cdot)$ | Cross-entropy loss |
| $D_{\mathrm{KL}}(\cdot||\cdot)$ | Kullbeck-Leibler divergence | $\mathcal{N}(\boldsymbol{\mu}, \boldsymbol{\Sigma})$ | Gaussian distribution |
| $\mathcal{U}\{a, b\}$ | Discrete uniform distribution | | |
| | **Linear Algebra** | | |
| $\boldsymbol{I}$ | Identity matrix | $\boldsymbol{V}$ | Codebook |
| $\boldsymbol{J}$ | Jordan normal form | $\boldsymbol{N}$ | Nilpotent matrix |
| $\boldsymbol{M}^k$ | Zeros matrix except $k^{th}$ column vector | $\boldsymbol{m}^k$ | $k^{th}$ column vector of $\boldsymbol{M}^k$ |
| $\boldsymbol{z}$ | Latent vector $\in \mathbb{R}^n$ | $\boldsymbol{z}^k$ | $k^{th}$ dimension value of $\boldsymbol{z}$ |
| $\boldsymbol{c}$ | Latent vector $\in \mathbb{C}^n$ | $\boldsymbol{c}^k$ | $k^{th}$ dimension value of $\boldsymbol{c}$ |
| $\boldsymbol{\theta}$ | $\boldsymbol{\theta} \in \Theta^n$ | $\boldsymbol{\theta}^k$ | $k^{th}$ dimension value of $\boldsymbol{\theta}$ |
| $\theta_c$ | Canonical point | $\vec{0}$ | Zeros vector |
| $\Delta \boldsymbol{v}$ | $\sum_k \Delta \boldsymbol{v}^k$ | $\Delta \boldsymbol{v}^k$ | Sparse vector ($k^{th}$ dimension value $\in \mathbb{R}\backslash\{0\}$) |
| $\boldsymbol{k}$ | Step | $\boldsymbol{k}_{i \to j}$ | Step from $\boldsymbol{\theta}_i$ to $\boldsymbol{\theta}_j$ |
| | **Others** | | |
| $\mathbb{R}$ | Real number | $\mathbb{Z}^+$ | Integer |
| $\mathbb{C}$ | Complex number | $\mathfrak{R}$ | Real part of complex number |
| $\mathfrak{I}$ | Imaginary part of complex number | $|F_i|$ | number of factors |
| $||\cdot||$ | L1 norm | $||\cdot||_2$ | L2 norm |
| $l$ | Ground truth | | |

## A  PROOF OF LIMITATIONS

We consider the dimension-wise disentangled representation, so we constraint a few conditions as follows:

**Condition A.1.** There exists an equivariant function $q_\phi \circ \Omega : \mathcal{F} \to \mathcal{Z}$ mapping fully disentangled factor and latent space.

**Condition A.2.** $\mathcal{Z}$ is a $G_z$-set that is a symmetry group acting on $Z$.

**Condition A.3.** Group element $g_z$ only affects to a single dimension value of latent vector $z$, where $g_z \in G_z$.

### A.1  PROOF: LIMITATIONS OF $GL(n)$

**Proof: Limitations of $GL(n)$, implemented by the matrix exponential, to Represent the Cyclic Semantics on the Disentangled Space**

**Condition A.4.** The symmetry group $G_z$ ($GL'(n)$) acting on latent vector space is defined as a subgroup of the General Linear group, implemented with matrix exponential.

**Condition A.5.** For $g^k \in \mathbb{R}^{n \times n}$ and $g = \prod_k g^k$, $g^k$ only affects the $k^{th}$ dimension value of vector $z$.

In prior works (Jung et al., 2024; Zhu et al., 2021; Kuzina et al., 2022; Miyato et al., 2022; Marchetti et al., 2023a), the General Linear group $GL(n)$ is usually implemented with the Lie algebra $\mathfrak{g}$ to represent the symmetries between two inputs in the latent vector space:

$$g = e^{\sum_i \alpha_i \mathfrak{g}_i}, \tag{7}$$

where $g \in GL(n)$, $\alpha \in \mathbb{R}$, $\mathfrak{g} \in \mathbb{R}^{n \times n}$, and the matrix exponential $e^{\mathfrak{g}}$ defined as $e^{\mathfrak{g}} = \sum_{n=0}^{\infty} \frac{1}{n!} \mathfrak{g}^n$. In addition, group $GL(n)$ acts on the latent vector space $\mathcal{Z}$ with group action:

$$\alpha(g, z) = gz, \tag{8}$$

where latent vector $z \in \mathbb{R}^n$, commonly used in previous works (Zhu et al., 2021; Jung et al., 2024; Kuzina et al., 2022; Marchetti et al., 2023a). We first show the property of disentangled representation with matrix exponential.

**Proposition A.6.** *Let the symmetry group $G_z$ ($GL'(n)$) is defined as a subgroup of the General Linear group that implemented with matrix exponential, where $GL'(n) = \{e^M | M \in \mathbb{R}^{n \times n}\}$, $g^k$ is an element of $GL'(n)$, and $g = \prod_k g^k$. Then $e^g z = eIgz + v'$.*

*Proof.* If group element $g$ acts on latent vector then, $gz - z = \Delta v$, where $\Delta v = \sum_k \Delta v^k$, $\Delta v^k$ is a sparse vector ($k^{th}$ dimension value $\in \mathbb{R} \setminus \{0\}$, otherwise it is zero), and $g^k z - z = \Delta v^k$. Then we define $(g)^n z - z = n \Delta v + (n-1) v'_n$, where $v'_n \in \mathbb{R}^n$ is an arbitrary real vector. Group element $g$ represents the cyclic semantics of the dataset space, then it satisfies the following equation:

$$
\begin{aligned}
(g_i - I)z &= \Delta v \\
\frac{1}{2!}((g_i)^2 - I)z &= \frac{2}{2!}(\Delta v + \frac{1}{2} v'_2) \\
&\vdots \\
\lim_{n \to \infty} \frac{1}{n!}((g_i)^n - I)z &= \lim_{n \to \infty} \frac{1}{(n-1)!}(\Delta v + \frac{1}{n} v'_n).
\end{aligned}
\tag{9}
$$

By adding left- and right-hand side of Eq. 9, we then get:

$$
\begin{aligned}
&\Rightarrow (e^{g_i} - I)z - (e-1)Iz = eI\Delta v + v' \\
&\Rightarrow e^{g_i} z = eIg_i z + v',
\end{aligned}
\tag{10}
$$

where $g_i \in G$, $v' = \lim_n \sum_n \frac{1}{n!} v'_n$ and $v' = v' + eI\Delta v$. $\square$

**Lemma A.7.** *By the Proposition A.6, if $v' = \vec{0}$ in Eq. 10, then $g = I$ and the index of the nilpotent matrix of Jordan normal form of $\mathfrak{g}$ is 2.*

*Proof.* The trivial solution of $(e^g - e\boldsymbol{I}g)\boldsymbol{z} = 0, \forall \boldsymbol{z} \in \mathcal{Z}$ is that

$$e^g - e\boldsymbol{I}g = 0. \tag{11}$$

Every matrix $\mathfrak{g} \in \mathbb{C}^{n \times n}$ has a Jordan normal form $\boldsymbol{J}$ as $\mathfrak{g} = \boldsymbol{S}\boldsymbol{J}\boldsymbol{S}^{-1}$. Then group element $(g = e^{\mathfrak{g}})$ follows as:

$$
\begin{aligned}
e^{\mathfrak{g}} &= \lim_{n\to\infty} \boldsymbol{I} + \boldsymbol{S}\boldsymbol{J}\boldsymbol{S}^{-1} + \frac{1}{2!}\boldsymbol{S}\boldsymbol{J}^2\boldsymbol{S}^{-1} + \cdots + \frac{1}{n!}\boldsymbol{S}\boldsymbol{J}^n\boldsymbol{S}^{-1} \\
&= \lim_{n\to\infty} \boldsymbol{I} + \boldsymbol{S}(\boldsymbol{J} + \frac{1}{2!}\boldsymbol{J}^2 + \cdots + \frac{1}{n!}\boldsymbol{J}^n)\boldsymbol{S}^{-1} \\
&= \boldsymbol{I} + \boldsymbol{S}(e^{\boldsymbol{J}} - \boldsymbol{I})\boldsymbol{S}^{-1} \\
&= \boldsymbol{S}e^{\boldsymbol{J}}\boldsymbol{S}^{-1}
\end{aligned}
\tag{12}
$$

In the same way, the exponential of $g$ is equal to:

$$e^g = \boldsymbol{S}e^{e^{\boldsymbol{J}}}\boldsymbol{S}^{-1} \tag{13}$$

Therefore group element $g$ satisfies

$$
\begin{aligned}
\boldsymbol{S}e^{e^{\boldsymbol{J}}}\boldsymbol{S}^{-1} &= e\boldsymbol{I}\boldsymbol{S}e^{\boldsymbol{J}}\boldsymbol{S}^{-1} \\
\Rightarrow e^{e^{\boldsymbol{J}}} &= e^{\boldsymbol{I}}e^{\boldsymbol{J}} \; (\because e\boldsymbol{I} = e^{\boldsymbol{I}}) \\
\Rightarrow e^{e^{\boldsymbol{J}}} &= e^{\boldsymbol{I}+\boldsymbol{J}} \; (\because \boldsymbol{I}\boldsymbol{J} = \boldsymbol{J}\boldsymbol{I}) \\
\therefore e^{\boldsymbol{J}} &= \boldsymbol{I} + \boldsymbol{J}
\end{aligned}
\tag{14}
$$

If $\boldsymbol{J} = \boldsymbol{0}$, then $g$ satisfies the Eq. 14 and $g = \boldsymbol{I}$. If $\boldsymbol{J} \neq \boldsymbol{0}$ then,

$$
e^{\boldsymbol{J}} = \begin{bmatrix} e^{\lambda_1} & e^{\boldsymbol{J}}_{1,2} & \cdots & e^{\boldsymbol{J}}_{1,n} \\ & e^{\lambda_2} & \cdots & e^{\boldsymbol{J}}_{2,n} \\ & & \ddots & \vdots \\ & & & e^{\lambda_n} \end{bmatrix}, \text{ and } \boldsymbol{I}+\boldsymbol{J} = \begin{bmatrix} \lambda_1+1 & (\boldsymbol{I}+\boldsymbol{J})_{1,2} & \cdots & (\boldsymbol{I}+\boldsymbol{J})_{1,n} \\ & \lambda_2+1 & \cdots & (\boldsymbol{I}+\boldsymbol{J})_{2,n} \\ & & \ddots & \vdots \\ & & & \lambda_n+1 \end{bmatrix}, \tag{15}
$$

where empty values are all zero. To satisify the Eq. 14, $\lambda_i = 0$ for $e_i^\lambda = \lambda_i + 1$, then $\boldsymbol{J} = \boldsymbol{D} + \boldsymbol{N} = \boldsymbol{N}$ because diagonal of $\boldsymbol{D}$ $\lambda_i = 0$, where $\boldsymbol{D}$ is a diagonal matrix and $\boldsymbol{N}$ is a nilpotent matrix. Therefore,

$$
\begin{aligned}
e^{\boldsymbol{J}} &= e^{\boldsymbol{N}} \text{ and } \boldsymbol{I} + \boldsymbol{J} = \boldsymbol{I} + \boldsymbol{N} \\
\Rightarrow e^{\boldsymbol{N}} &= \boldsymbol{I} + \boldsymbol{N} \\
\Rightarrow \lim_{n\to\infty} \frac{1}{2!}\boldsymbol{N}^2 &+ \frac{1}{3!}\boldsymbol{N}^3 + \cdots + \frac{1}{n!}\boldsymbol{N}^n = 0
\end{aligned}
\tag{16}
$$

Therefore if the index of nilpotent matrix is 2 and $e_{i,j}^{\boldsymbol{J}} = (\boldsymbol{I}+\boldsymbol{J})_{i,j}$ where $i < j$, then it satisfies the Eq. 11

$\square$

**Lemma A.8.** *If the index of the nilpotent matrix of Jordan normal form of $\mathfrak{g}$ is 2, then $g = e^{\mathfrak{g}}$ does not represent the cyclic semantics.*

*Proof.* If $g$ is an element of cyclic group then there exists $n^{th}$ power of $g$ such that $g^n$ is the identity matrix.

$$
\begin{aligned}
g^n &= \boldsymbol{S}(\boldsymbol{I} + n\boldsymbol{N})\boldsymbol{S}^{-1} = \boldsymbol{I} \; (\because \boldsymbol{N}^2 = \boldsymbol{0}) \\
\Rightarrow \boldsymbol{I} + \boldsymbol{S}n\boldsymbol{N}\boldsymbol{S}^{-1} &= \boldsymbol{I} \\
\Rightarrow \boldsymbol{S}n\boldsymbol{N}\boldsymbol{S}^{-1} &= \boldsymbol{0}.
\end{aligned}
\tag{17}
$$

To satisfy the Eq. 17, $\boldsymbol{N} = \boldsymbol{0}$ because the index of $\boldsymbol{N}$ is 2. $\square$

By the Lemma A.7 and A.8, there exists only one element to represent the cyclic semantics of the dataset in the disentangled space.

**Lemma A.9.** *If $v' \in \mathbb{R}^n \backslash \{0\}$, then 1) $g^k$ represents the cyclic semantic as $g^k = I + M^k$ and $m^k \in \mathbb{R}^n$, where $m^k$ is a $k^{th}$ column vector of $M^k$, $m^j = 0$ and $j \in \{1, 2, \ldots, n\} \backslash \{k\}$.*

*Proof.* As we define that $g^k$ only affect to a single dimension value of $z$ we rewirte Eq. 10 as follows:

$$e^{g^k} z = e I g^k z + v'^k, \tag{18}$$

where $v'^k$ is a sparse vector. For Eq. 18 $\forall z \in \mathcal{Z}$ then symmetry $g^k$ follows as:

$$e^{g^k} - e I g^k = \begin{bmatrix} \vec{0} & \cdots & \vec{0} \, m^k \, \vec{0} \cdots \vec{0} \end{bmatrix}, \tag{19}$$

where $\vec{0}$ is a zero column vector.

Then, satisfying the Eq. 19 and affecting a single dimension:

$$\begin{aligned}
z^\mathsf{T} g^k &= [z_1, \cdots, z_{k-1}, z_k + \alpha, z_{k+1}, \cdots, z_n] \\
z^\mathsf{T} e I g^k &= [ez_1, \cdots, ez_{k-1}, e(z_k + \alpha), ez_{k+1}, \cdots, ez_n] \\
\therefore z^\mathsf{T} e^{g^k} &= [ez_1, \cdots, ez_{k-1}, z_k + \beta, ez_{k+1}, \cdots, ez_n] \\
\Rightarrow z^\mathsf{T}(e^{g^k} - eI) &= [0, \cdots, 0, (1-e)z_k + \beta, \cdots, 0].
\end{aligned} \tag{20}$$

For Eq. 20 for all $z$, then

$$\begin{aligned}
e^{g^k} - eI &= [\vec{0} \cdots \vec{0} \, m'^k \, \vec{0} \cdots \vec{0}] \\
\therefore e^{g^k} &= eI + [\vec{0} \cdots \vec{0} \, m'^k \, \vec{0} \cdots \vec{0}].
\end{aligned} \tag{21}$$

By the same way, $g^k = I + [\vec{0} \cdots \vec{0} \, m^k \, \vec{0} \cdots \vec{0}] = I + M^k$ because $z^\mathsf{T}(g^k - I)$ is a sparse vector. $\square$

**Lemma A.10.** *For $g_k$ to represent cyclic semantics, $g_k = I$.*

*Proof.* If $I + M^k$ represents the cyclic semantic the, there exists a $n$ where $(I + M^k)^{(n+1)} = I$. The fomation of the power of the matrix $(I + M^k)^n$ is also $I + M^k_n$, where $M^k_n \in M^k$ is an arbitrary real matrix. Therefore $M^k_n = 0$, then $g^k = I$. $\square$

It implies that $g^k$ represents only an identity transformation of dataset space.

**Theorem A.11.** *(Limit of $GL'(n)$) According to Proposition A.6, only the identity matrix $(g = I)$ represents the cyclic semantics of the dataset with consistent symmetries, where $g \in GL'(n)$*

*Proof.* Through the Lemma A.7 to Lemma A.10, if the group element $g$ represents the cyclic semantics, then only the identity matrix satisfies the Eq. 9. There is always a case as $g = e^{\mathfrak{g}}$, and $(g)^{n-1} = (e^{\mathfrak{g}})^{n-1}$ but $(g)^n \neq (e^{\mathfrak{g}})^n$, where $g \neq I$, because $g$ can not represent the cyclic semantic. By the equivariant function $q_\phi$:

$$\begin{aligned}
q_\phi(x_1) &= z_1 \\
q_\phi(g_x \circ_x x_1) &= \Gamma(g_x) \circ_z z_1 \\
&\vdots \\
q_\phi(g_x^{n-1} \circ_x x_1) &= [\Gamma(g_x)]^{n-1} \circ_c z_1 \\
q_\phi(g_x^n \circ_x x_1) &\neq [\Gamma(g_x)]^n \circ_c z_1 \; (\because [\Gamma(g_x)]^n \neq (e^{\mathfrak{g}})^n).
\end{aligned} \tag{22}$$

It implies that cyclic semantic $g_x$ between the $x_k$ and $x_{k+1}$ is divided as:

$$\Gamma(g_x) = \begin{cases} e^{\mathfrak{g}} & \text{if } k < n \\ e^{\mathfrak{g}'} & \text{if } k = n \end{cases}, \text{ where } \mathfrak{g} \neq \mathfrak{g}'. \tag{23}$$

$\square$

Therefore, representing the cyclic semantics of the dataset with consistent symmetry is impossible according to the Theorem A.11.

**Limitations of $GL(n)$ to Represent the Cyclic Semantics on the Disentangled Space**

**Theorem A.12.** *(Limit of GL(n)) If $H \subset GL(n)$, then only the identity matrix of $GL(n)$ represents the cyclic semantics of the dataset with consistent symmetries, where $H = \{h | h = \boldsymbol{I} + \boldsymbol{M}^k\}$, $\boldsymbol{m}^k$ is a column vector of $\boldsymbol{M}^k$, $\boldsymbol{m}^j = \vec{0}$ and $j \in \{1, 2, \ldots, n\} \backslash \{k\}$.*

*Proof.* If the invertible matrix $g$ changes a single dimension value of the latent vector $\boldsymbol{z}$ then,

$$
\begin{aligned}
\boldsymbol{z}^\mathsf{T} g - \boldsymbol{z} &= [0, \ldots, 0, \alpha, 0, \ldots, 0] \\
\Rightarrow \boldsymbol{z}^\mathsf{T}(g - \boldsymbol{I}) &= [0, \ldots, 0, \alpha, 0, \ldots, 0].
\end{aligned}
$$

$$
\Rightarrow [z_1, \ldots, z_n]
\begin{bmatrix}
g_{11} - 1 & \cdots & g_{1n} \\
\vdots & \ddots & \vdots \\
g_{n1} & \cdots & g_{nn} - 1
\end{bmatrix}
= [0, \ldots, 0, \alpha, 0, \ldots, 0]
\tag{24}
$$

As defined the $G$-set as a latent vector space $\mathcal{Z}$, group element $g$ satisfies the Eq. 24 over all vectors. Then $g - \boldsymbol{I}$ elements are all 0, except the $k^{th}$ column vector ($\boldsymbol{m}^k \neq 0$). Therefore, group element $g \in H$. $\square$

**Theorem A.13.** *If $h \in H \backslash \{\boldsymbol{I}\}$, then this set does not represent the consistent symmetry with cyclic semantics.*

*Proof.* According to Lemma A.10, $h^k = \boldsymbol{I}$. Therefore, $h^k$ represents only the identity transformation of the dataset space. $\square$

Therefore, representing the cyclic semantics of the dataset with consistent symmetry implemented by the $GL(n)$ is impossible except in the case of the identity matrix.

### A.2 PROOF: LIMITATIONS OF VECTOR ADDITION FOR CYCLIC SEMANTICS ON THE DISENTANGLED SPACE

In the previous works (Hwang et al., 2023), the vector addition is used to define a group action between two latent vectors for an equivariant model, where the group action $\alpha(g, \boldsymbol{z}) = g + \boldsymbol{z}$.

**Theorem A.14.** *If the group action is defined as $\alpha(g, \boldsymbol{z}_i) = g + \boldsymbol{z}_i$, then $g$ does not represent the consistent symmetry for cyclic semantics with disentangled representation, where $g \in G \backslash \{\vec{0}\}$ and $\boldsymbol{z} \in \mathbb{R}^n$.*

*Proof.* If $g$ represents the cyclic semantics of $\mathcal{X}$, then there exists:

$$
(g)^n = ng = \vec{0}.
\tag{25}
$$

The solution of Eq. 25 is $g = \vec{0}$. There is always a case as $g = \boldsymbol{a}$, and $(g)^{n-1} = (n-1)\boldsymbol{a}$ but $(g)^n \neq n\boldsymbol{a}$, where $g \neq \vec{0}$, because $g$ can not represent the cyclic semantics. By the equivariant function $q_\phi$:

$$
\begin{aligned}
q_\phi(x_1) &= z_1 \\
q_\phi(g_x \circ_x x_1) &= \Gamma(g_x) \circ_z z_1 \\
&\vdots \\
q_\phi(g_x^{n-1} \circ_x x_1) &= (n-1)[\Gamma(g_x)] \circ_c z_1 \\
q_\phi(g_x^n \circ_x x_1) &\neq n[\Gamma(g_x)] \circ_c z_1 \ (\because n[\Gamma(g_x)] \neq n\boldsymbol{a}).
\end{aligned}
\tag{26}
$$

It implies that cyclic semantic $g_x$ between the $x_k$ and $x_{k+1}$ is divided as:

$$
\Gamma(g_x) = \begin{cases} \boldsymbol{a} \text{ if } k < n \\ \boldsymbol{a}' \text{ if } k = n \end{cases} \quad , \text{ where } \boldsymbol{a} \neq \boldsymbol{a}'.
\tag{27}
$$

$\square$

According to Theorem A.14, the group action defined as $\alpha(g, \boldsymbol{z}) = g + \boldsymbol{z}$ represents the cyclic semantics of the dataset with consistent symmetry when $g = \vec{0}$. However, the group elements are insufficient to encompass the entire cyclic semantics of input space. Additionally, this causes the inconsistency issue when holding consistency with vector addition.

### A.3 Proof: Loss of Symmetry Structure with Endomorphism

**Definition A.15.** Let $(G, \cdot), (H, \circ)$ be two groups. If mapping function $\Gamma : G \to H$, *s.t.* $\Gamma(g_i \cdot g_j) = \Gamma(g_i) \circ \Gamma(g_j)$, then $\Gamma$ is called homomorphism.

**Definition A.16.** Let $\Gamma$ is surjective then $\Gamma$ is called endomorphism.

**Definition A.17.** Let a special case of homomorphism where $\Gamma$ is bijective is called isomorphism.

**Corollary A.18.** *By the equivariant and surjective function* $b : \mathcal{Z} \to \mathcal{Y}$*, the capacity of* $\mathcal{Z}$ *and* $\mathcal{Y}$ *is* $|\mathcal{Z}| \geq |\mathcal{Y}|$ *then* $\Gamma$ *is an endomorphism because* $|G_{\mathcal{Z}}| \geq |G_{\mathcal{Y}}|$*. On the other hand, isomorphism identically maps the two spaces (*$|G_{\mathcal{Z}}| = |G_{\mathcal{Y}}|$*).*

Therefore, if $b$ is surjective and not injective, then there exists at least one case where $\Gamma'(g_i) = \Gamma'(g_j)$. It implies that loss of symmetry structure occurs with a surjective function.

## B Details of Experiments Setting

### B.1 Resources

We set the following settings for all experiments on a single Galaxy 2080Ti GPU, a single Galaxy 3090 GPU, and a single NVIDIA A100 GPU for the dSprites 3D Shapes and MPI3D datasets. The Python version is 3.7.10, and the PyTorch version is 1.9.1. More details are in the README.md file.

### B.2 Datasets

1) The dSprites dataset consists of 737,280 binary $64 \times 64$ images with five independent ground truth factors(number of values), *i.e.* x-position (32), y-position (32), orientation (40), shape (3), and scale (6) (Matthey et al., 2017). Any composite transformation of x- and y-position, orientation (2D rotation), scale, and shape is commutative. 2) The 3D Shapes dataset consists of 480,000 RGB $64 \times 64 \times 3$ images with six independent ground truth factors: orientation (15), shape (4), floor color (10), scale (8), object color (10), and wall color (10) (Burgess & Kim, 2018). 3) The MPI3D (real-world complex) dataset consists of 460,800 RGB $64 \times 64 \times 3$ images with seven independent ground truth factors: color (4) shape (4), size (2), height (3), background color (3), horizontal (40), and vertical axis (40) (Gondal et al., 2019). Additionally, we use cLPR dataset[1] consists of 250,047 RGB $64 \times 64 \times 3$ images with three independent ground truth factors: x-rotation (63), y-rotation (63), and z-rotation (63).

### B.3 Setting for Combinatorial Generalization

**Train and Test datasets**  We except the case [shape=$ellips$, position-x $\geq 0.6$, position-y $\geq 0.6, 120° \leq$ rotation $\leq 240°$, scale $< 0.6$] for dSprites r2e training set and [shape=$square$, position-x $\geq 0.5$] for dSprites r2r training set.

We except the case [floor-hue $> 0.5$, wall-hue $> 0.5$, object-hue $\geq 0.5$, shape=cylinder, object-scale=1, object-orientation=0] for 3D Shapes r2e training set and [object-hue $\geq 0.5$, shape=oblong] for 3D Shapes r2r training set.

We except the case [shape $=$ cone, object size $= 0$, cameraheight $= 1$, background color $=$ purple, object color $\in \{$blue, brown, olive$\}$, horizontal axis $\geq 20$, vertical axis $\geq 20$]for r2e training set and [shape $=$ cylinder, scale $< 6$, orientation, $16 \leq$ horizontal axis $< 32$, vertical axis] for r2r training set.

---

[1]https://github.com/yvan/cLPR

**Hyper-Parameter Tuning** We set $\beta \in \{1, 2, 10\}$ for $\beta$-VAE (Higgins et al., 2017) and $\beta$-TCVAE (Chen et al., 2018), and $\beta \in \{1, 10\}$ for VAE-MAGA, which employs the MAGA-net proposed module on the CNN-based encoder and decoder, instead of Glow (Kingma & Dhariwal, 2018). We set the common hyper-parameters of the proposed method at $\alpha \in \{100, 1000\}$, $\gamma \in \{1, 10\}$, $\lambda \in \{0.0, 1.0\}$ for the supervised and ground truth models, and $\beta \in \{1.0, 2.0\}$ for the supervised method. We run each model with three seeds, $\in \{1, 2, 3\}$. We set $N = 1000$ and $n' = 10$.

**Decoder Equivariant Loss** For combinatorial generalization, we add the decoder equivariant loss as:

$$\mathcal{L}_{de} = R.E(x_j, p_\theta \circ g_{i \to j} \circ_\theta (f' \circ f \circ q_\phi(x_i))), \tag{28}$$

where $R.E(\cdot)$ is a reconstruction error and $p_\theta$ is a decoder. We add the $\mathcal{L}_{de}$ to the objective losses with hyper-parameter $\lambda$.

### B.4 SETTING FOR DISENTANGLEMENT LEARNING

**Hyper-Parameter Tuning** We set $\beta \in \{1, 2, 10\}$ for $\beta$-VAE (Higgins et al., 2017) and $\beta$-TCVAE (Chen et al., 2018), $\gamma \in \{10, 20, 40\}$ for Factor-VAE (Kim & Mnih, 2018), $hy_{rec} \in \{0.1, 0.2, 0.7\}$ for CLG-VAE (Zhu et al., 2021), $\beta = 1$ for Ada-GVAE (Locatello et al., 2020). We set the common hyper-parameters of proposed method $\alpha \in \{100, 1000\}$, $\gamma = 1$ for superivsed and ground truth model, $\beta \in \{1.0, 2.0\}$ for supervised method, and $\lambda = 1.0, p = 0.5$ for semi-supervised method. We run 10 seed variance over each model with seed $\in \{1, 2, \ldots, 10\}$. We set $N = 1000$ and $n' = 10$. We evaluate four metrics $\beta$-VAE metric (Higgins et al., 2017), Factor VAE metric (Kim & Mnih, 2018), SAP (Kumar et al., 2018), and DCI (Eastwood & Williams, 2018).

## C  COMBINATORIAL GENERALIZATION RESULTS

As shown in Fig. 9a, we selected the four worst samples (those with the highest reconstruction errors): 1) $\beta$-VAE results contain only position semantics, 2) $\beta$-TCVAE captures the position and scale values but fails to capture the shape and rotation factors, 3) VAE-MAGA struggles with generalization. Even though our method does not capture all semantics, it shows improvement compared to the baselines: 4) the supervised method misses either the shape or rotation, and 5) the GT model only misses the shape semantic. As shown in Fig. 9b, the representations of the baseline are not close to a disentangled representation. In contrast, the representation of the supervised method approaches a disentangled representation and shows better generalization. This implies that a disentangled representation containing the symmetry structure could benefit combinatorial generalization.

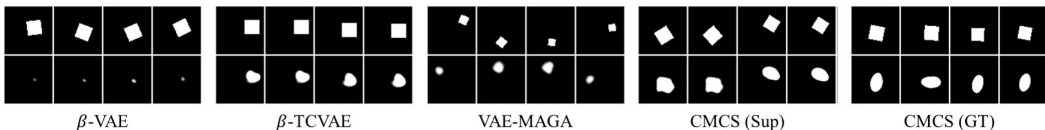

$\beta$-VAE          $\beta$-TCVAE          VAE-MAGA          CMCS (Sup)          CMCS (GT)

(a) Visualization of generated images of the worst 4 samples of R2R task. Each $1^{st}$ row is a group truth, and $2^{nd}$ row is a generated image. The red box indicates the results, which do not contain all the semantics of ground truth.

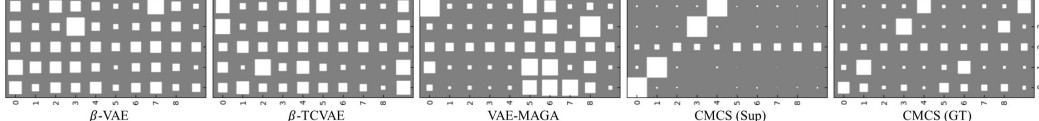

$\beta$-VAE          $\beta$-TCVAE          VAE-MAGA          CMCS (Sup)          CMCS (GT)

(b) Visualization of DCI metric of dSprites dataset

Figure 9: Qualitative results of combinatorial generalization of dSprites dataset (R2R). A more sparse matrix implies clear disentanglement.

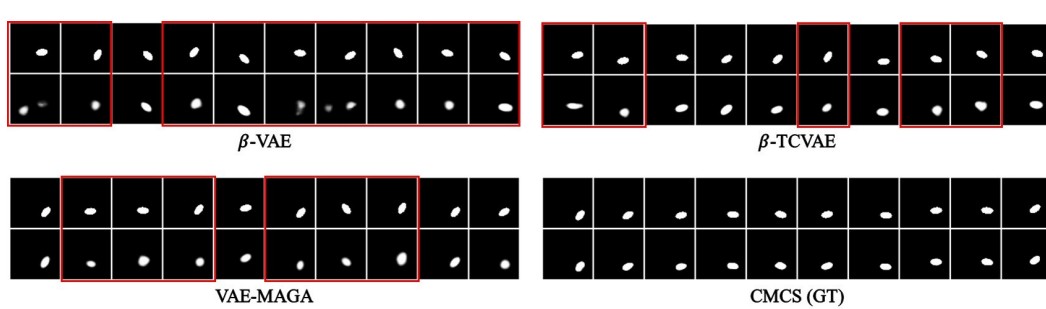

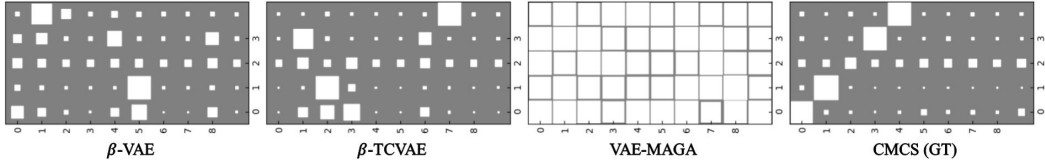

(a) Visualization of generated images of the worst 4 samples of R2E task. Each $1^{st}$ row is a group truth, and $2^{nd}$ row is a generated image. The red box indicates the results, which do not contain all the semantics of ground truth.

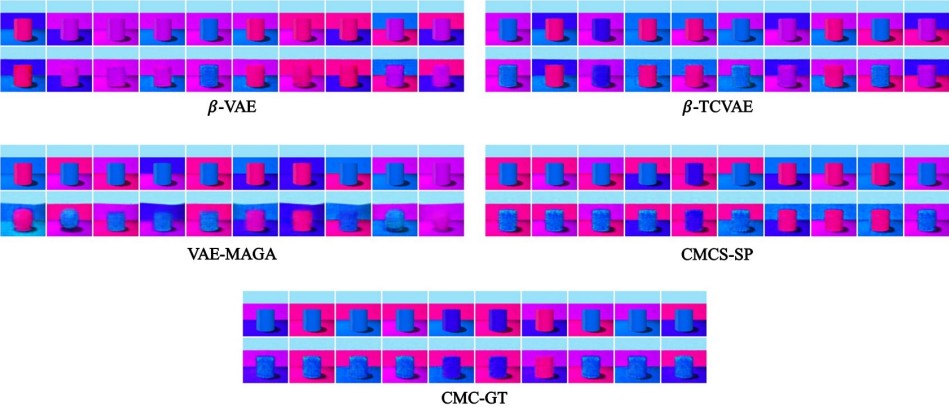

(b) Visualization of DCI metric of dSprites dataset

Figure 10: Qualitative results of combinatorial generalization of dSprites dataset (R2E). A more sparse matrix implies clear disentanglement.

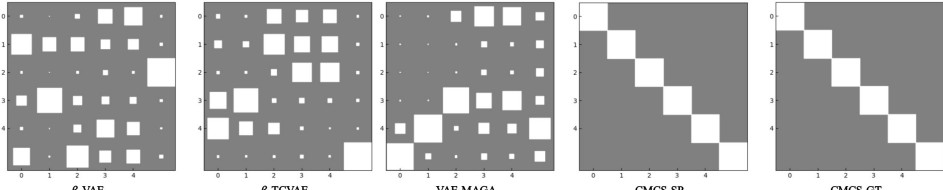

(a) Visualization of generated images of the worst 10 samples of the R2R task. Each $1^{st}$ and $2^{nd}$ row shows the group truth samples and the generated results, respectively. The red box indicates the negative results, which do not contain all the semantics of the ground truth. We utilize randomly selected pivot images as introduced in Hwang et al. (2023) for the CMCS model.

(b) Visualization of DCI metric of 3D Shapes dataset (training set). A more sparse matrix implies clear disentanglement.

Figure 11: Qualitative results of combinatorial generalization of 3D Shapes dataset (R2E).

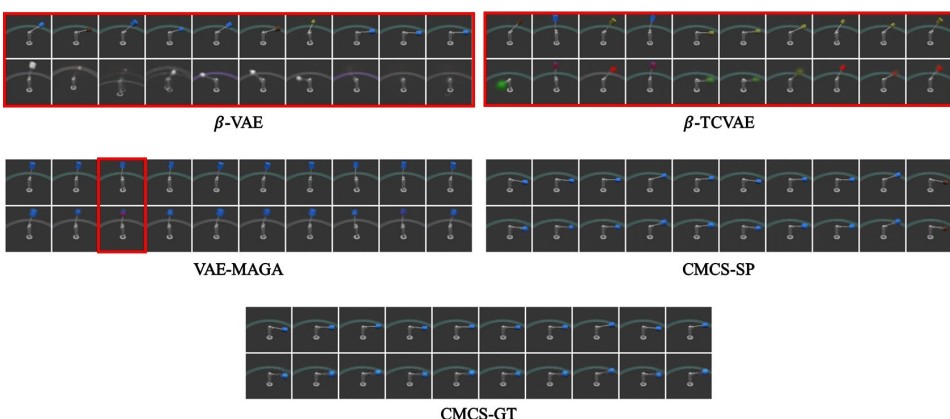



(a) Visualization of generated images of the worst 10 samples of the R2R task. Each $1^{st}$ and $2^{nd}$ row shows the group truth samples and the generated results, respectively. The red box indicates the negative results, which do not contain all the semantics of the ground truth. We utilize randomly selected pivot images as introduced in Hwang et al. (2023) for the CMCS model.

(b) Visualization of DCI metric of 3D Shapes dataset (training set). A more sparse matrix implies clear disentanglement.

Figure 12: Qualitative results of combinatorial generalization of MPI3D dataset (R2E).

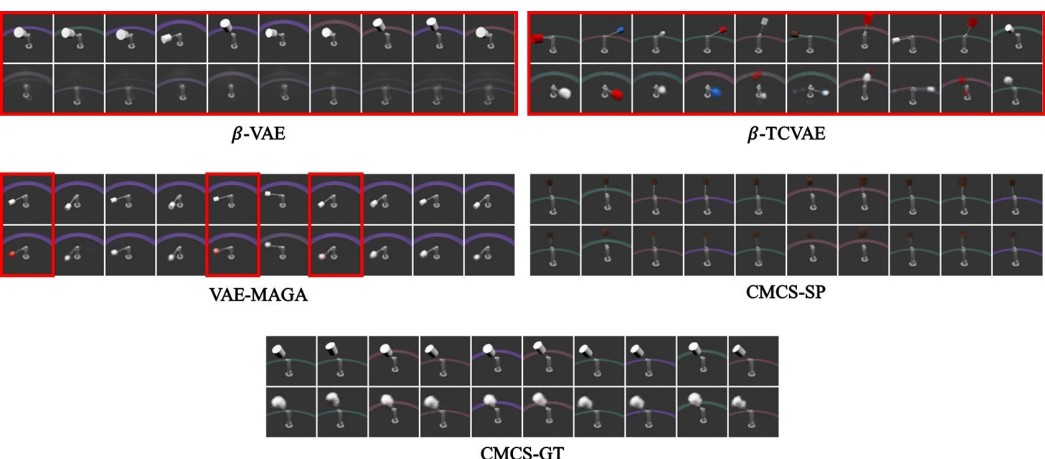



(a) Visualization of generated images of the worst 10 samples of the R2R task. Each $1^{st}$ and $2^{nd}$ row shows the group truth samples and the generated results, respectively. The red box indicates the negative results, which do not contain all the semantics of the ground truth. We utilize randomly selected pivot images as introduced in Hwang et al. (2023) for the CMCS model.

(b) Visualization of DCI metric of 3D Shapes dataset (training set). A more sparse matrix implies clear disentanglement.

Figure 13: Qualitative results of combinatorial generalization of MPI3D dataset (R2R).

# D DISENTANGLEMENT LEARNING RESULTS

## D.1 TRADE-OFF (3D SHAPES)

As illustrated in Fig. 14, the proposed models improve the reconstruction error and disentanglement performance simultaneously on the dSprites dataset. Additionally, while the reconstruction error slightly decreases, the model performance dramatically improves compared to the baselines on the 3D Shapes dataset.

## D.2 GENERALIZATION WITH DISENTANGLED REPRESENTATIONS

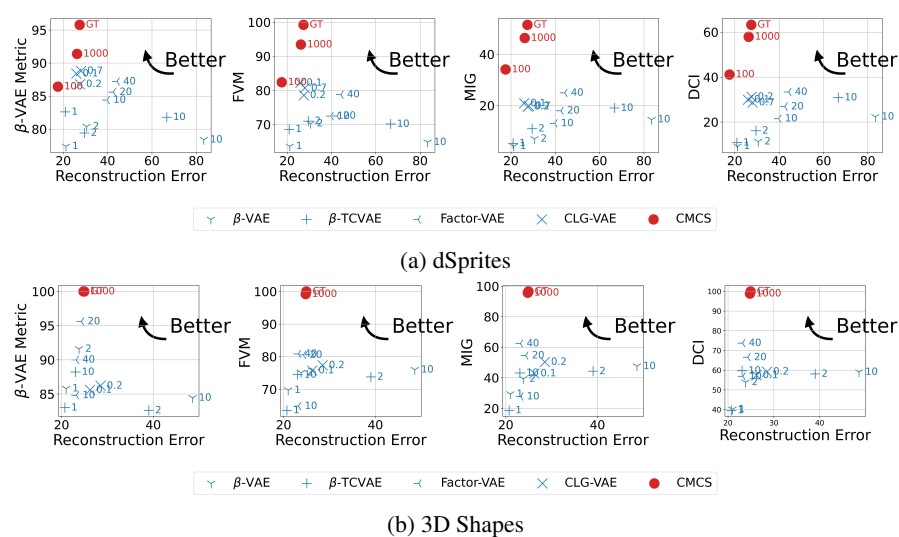

Figure 14: Reconstruction error vs. evaluation metrics ($\beta$-VAE metric, FVM, MIG, and DCI). The top left side indicates the best results for both objectives.

## D.3 IMPACT OF EACH LOSS

Table 5: Disentanglement performance over hyper-parameters

| $(\alpha, \gamma)$ | reconst. err. | beta-VAE | FVM | MIG | DCI |
|---|---|---|---|---|---|
| (100, 1) | **17.88**($\pm$1.24) | 88.40($\pm$4.97) | 82.13($\pm$1.86) | 33.88($\pm$3.03) | 42.27($\pm$2.02) |
| (200, 1) | 21.23($\pm$0.89) | 92.44($\pm$5.81) | 87.54($\pm$4.13) | 35.21($\pm$2.17) | 47.40($\pm$2.14) |
| (500, 1) | 26.00($\pm$2.72) | 94.00($\pm$4.42) | 96.41($\pm$1.83) | 43.73($\pm$3.55) | 55.36($\pm$2.70) |
| (1000, 1) | 27.41($\pm$0.73) | **95.80**($\pm$4.57) | **99.26**($\pm$1.12) | **51.81**($\pm$2.97) | **63.26**($\pm$2.73) |

(a) dSrites with Ground Truth model

| $(\alpha, \beta, \gamma)$ | reconst. err. | beta-VAE | FVM | MIG | DCI |
|---|---|---|---|---|---|
| (100, 1, 1) | **19.53**($\pm$1.75) | 85.78($\pm$5.87) | 82.10($\pm$2.81) | 27.12($\pm$1.98) | 34.42($\pm$1.04) |
| (100, 2, 1) | 17.58($\pm$1.08) | 86.44($\pm$5.90) | 82.40($\pm$1.62) | 34.08($\pm$1.75) | 41.16($\pm$0.98) |
| (1000, 1, 1) | 26.22($\pm$1.31) | **91.40**($\pm$4.90) | 93.46($\pm$1.97) | 46.35($\pm$1.94) | 57.92($\pm$1.97) |
| (1000, 2, 1) | 26.32($\pm$2.20) | **91.40**($\pm$4.99) | **93.74**($\pm$1.82) | **51.03**($\pm$2.42) | **64.69**($\pm$1.55) |

(b) dSrites with Supervised method

| $(\alpha, \gamma)$ | reconst. err. | beta-VAE | FVM | MIG | DCI |
|---|---|---|---|---|---|
| (100, 1) | 33.62($\pm$5.38) | 89.60($\pm$6.17) | 82.44($\pm$3.78) | 53.37($\pm$12.87) | 60.04($\pm$13.98) |
| (200, 1) | 34.50($\pm$5.35) | 95.00($\pm$5.34) | 91.95($\pm$7.14) | 68.06($\pm$17.82) | 74.16($\pm$14.69) |
| (500, 1) | 29.30($\pm$1.72) | **100.00**($\pm$0.00) | 97.28($\pm$3.05) | 86.68($\pm$5.56) | 90.68($\pm$5.87) |
| (1000, 1) | **24.94**($\pm$1.51) | **100.00**($\pm$0.00) | **100.00**($\pm$0.00) | **95.57**($\pm$0.80) | **99.94**($\pm$0.18) |

(c) 3D Shapes with Ground Truth model

Table 6: Disentanglement performance of Homomorphism VAE vs. Ours with dSprites.

| Model | beta-VAE | FVM | MIG | DCI |
|---|---|---|---|---|
| Homomorphism VAE | 18.80($\pm$5.75) | 30.24($\pm$12.18) | 0.39($\pm$0.76) | 1.35($\pm$1.12) |
| Groupified-VAE | 79.30($\pm$9.23) | 69.75($\pm$13.66) | 21.03($\pm$9.20) | 31.08($\pm$10.87) |
| CMCS-SP | **91.40**($\pm$4.99) | **93.74**($\pm$1.82) | **51.02**($\pm$2.42) | **64.69**($\pm$1.55) |

### D.4 OURS VS. HOMOMORPHISM VAE

As shown in Table 6, our model outperforms Homomorphism VAE (Keurti et al., 2023) and Groupified-VAE (Yang et al., 2021). Homomorphism VAE (Keurti et al., 2023) utilizes s the SO(2) for disentangled representation, and the elements of SO(2) affect the multi-dimension value of the latent vector. It implies that the SO(2) group is not appropriate symmetry for dimension-wise disentangled representation.

### D.5 IMPACT OF KNOWN LABEL RATIO

We set the known label ratio $p \in \{0.1, 0.2, 0.4, 0.5\}$. As shown in Table 7, our model is robust to the known label ratio except for the DCI metric.

Table 7: Disentanglement performance of semi-supervised learning methods.

| Model | dSprites | | | | 3DShapes | | | |
|---|---|---|---|---|---|---|---|---|
| | beta-VAE | FVM | MIG | DCI | beta-VAE | FVM | MIG | DCI |
| Ada-GVAE | 83.60($\pm$2.61) | 83.67($\pm$2.97) | 21.34($\pm$5.35) | 47.26($\pm$1.89) | 72.75($\pm$6.50) | 59.81($\pm$6.14) | 24.77($\pm$7.48) | 64.57($\pm$4.04) |
| CMCS-semi (0.1) | 88.60($\pm$7.72) | 83.36($\pm$3.51) | 14.71($\pm$1.25) | 23.46($\pm$1.69) | 86.80($\pm$3.90) | 84.05($\pm$2.66) | 55.17($\pm$2.18) | 61.79($\pm$3.15) |
| CMCS-semi (0.2) | 89.78($\pm$6.67) | 83.88($\pm$2.76) | 23.03($\pm$2.54) | 28.07($\pm$1.40) | 85.00($\pm$13.11) | 83.00($\pm$7.29) | 54.20($\pm$13.35) | 60.26($\pm$12.00) |
| CMCS-semi (0.4) | 88.20($\pm$5.92) | 83.49($\pm$2.22) | 31.87($\pm$2.02) | 39.44($\pm$0.79) | 86.40($\pm$6.98) | 87.61($\pm$7.09) | 61.47($\pm$9.51) | 67.87($\pm$8.39) |
| CMCS-semi (0.5) | 87.00($\pm$7.07) | 84.50($\pm$1.41) | 31.95($\pm$2.40) | 39.36($\pm$1.49) | 95.00($\pm$7.07) | 88.81($\pm$13.17) | 57.94($\pm$16.52) | 72.14($\pm$3.23) |

## E QUALITATIVE ANALYSIS OF DISENTANGLEMENT LEARNING

**3D Shapes** As shown in Fig. 15, the baseline results show that multiple factors are changed when a single dimension value is changed. On the other hand, ours show the fully disentangled results represent: $1^{st}$ row is the floor color changes, $2^{nd}$ row is the wall color changes, $3^{rd}$ row is the object color changes, $4^{th}$ row is the scale of object, $5^{th}$ row is the object shape changes, and $6^{th}$ row is the rotation changes.

**MPI3D** As shown in Fig. 16, the baseline results show that multiple factors are changed when a single dimension value is changed. Also, the object usually disappeared following the intervals with baselines. On the other hand, supervised methods show better results than baselines: $1^{st}$ row is the object color changes, $2^{nd}$ row is the object shape changes, $3^{rd}$ row is the object size changes, $5^{th}$ row is the background color changes, $7^{th}$ row is the vertical axis changes, and $9^{th}$ row is the height changes. Also, the GT model results represent: $1^{st}$ row is the object color changes, $2^{nd}$ row is the object shape changes, $3^{rd}$ row is the object size changes, $4^{th}$ row is the height changes, $5^{th}$ row is the background color changes, $6^{th}$ row is the vertical axis changes, and $7^{th}$ row is the horizontal axis changes.

**cLPR** As shown in Fig. 17, the Homomorphism VAE (Keurti et al., 2023) shows that multiple factors are changed when a single dimension value is changed. Also, the reconstruction quality is lower than ours (CMCS-GT and CMCS-SP). On the other hand, the supervised method shows better results than the Homomorphism VAE: $1^{st}$ and $3^{rd}$ rows are z-axis rotation, $2^{nd}$ row is y-zis rotation, and $4^{th}$ row is x-axis rotation. Also, the GT model results represent: $1^{st}$ and $2^{nd}$ rows are x-axis rotation, $3^{rd}$ row is z-axis rotation, and $4^{th}$ and $6^{th}$ rows are y-axis rotation.

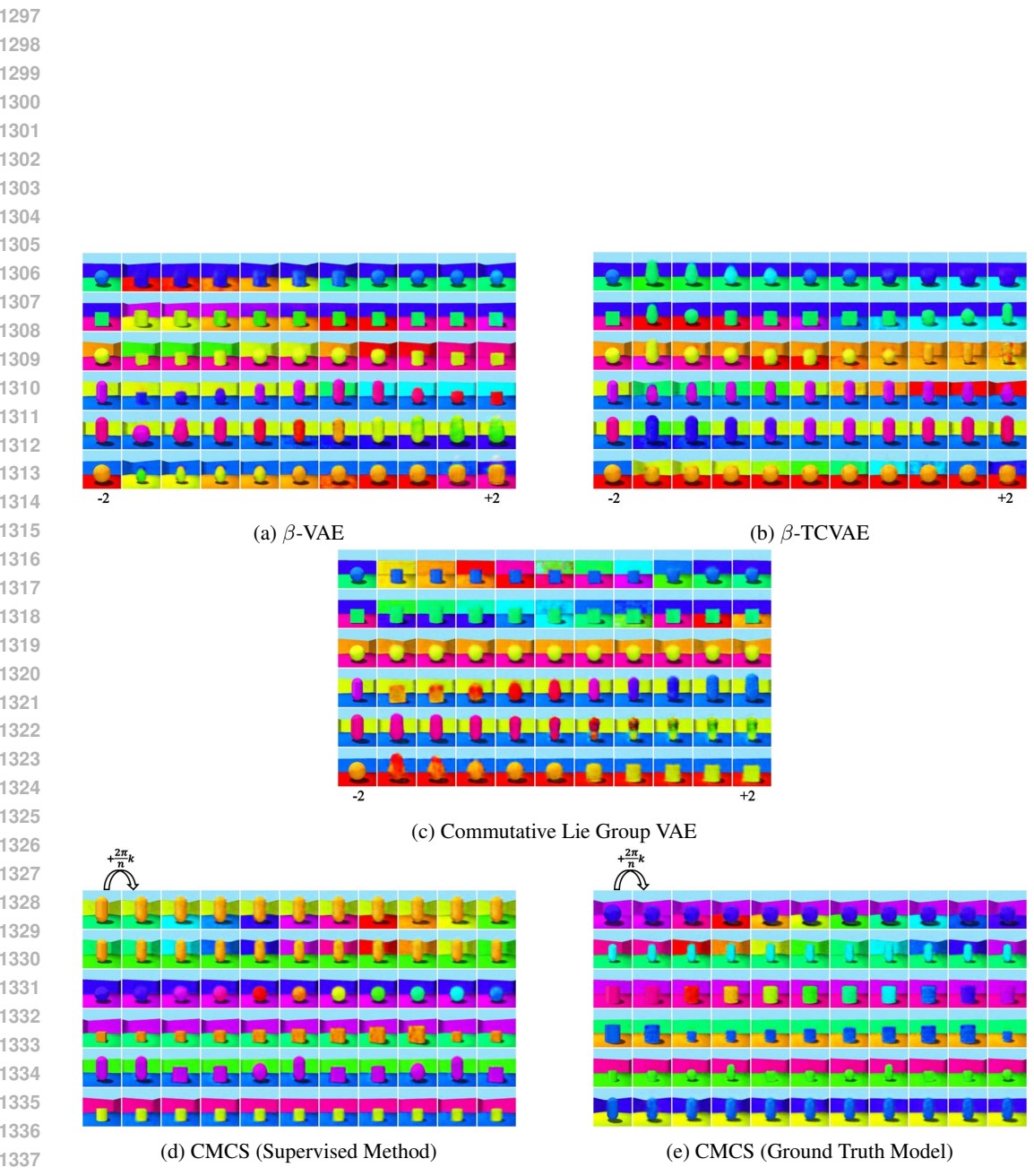

Figure 15: The $1^{st}$ column images are randomly selected from the dataset. Each row indicates each dimension of each model. $\beta$-VAE, $\beta$-TCVAE, and Commutative Lie Group VAE trace each dimension value from -2 to +2. The proposed methods apply a group action $+\frac{2\pi}{n}$ to the selected images a total of 10 times.

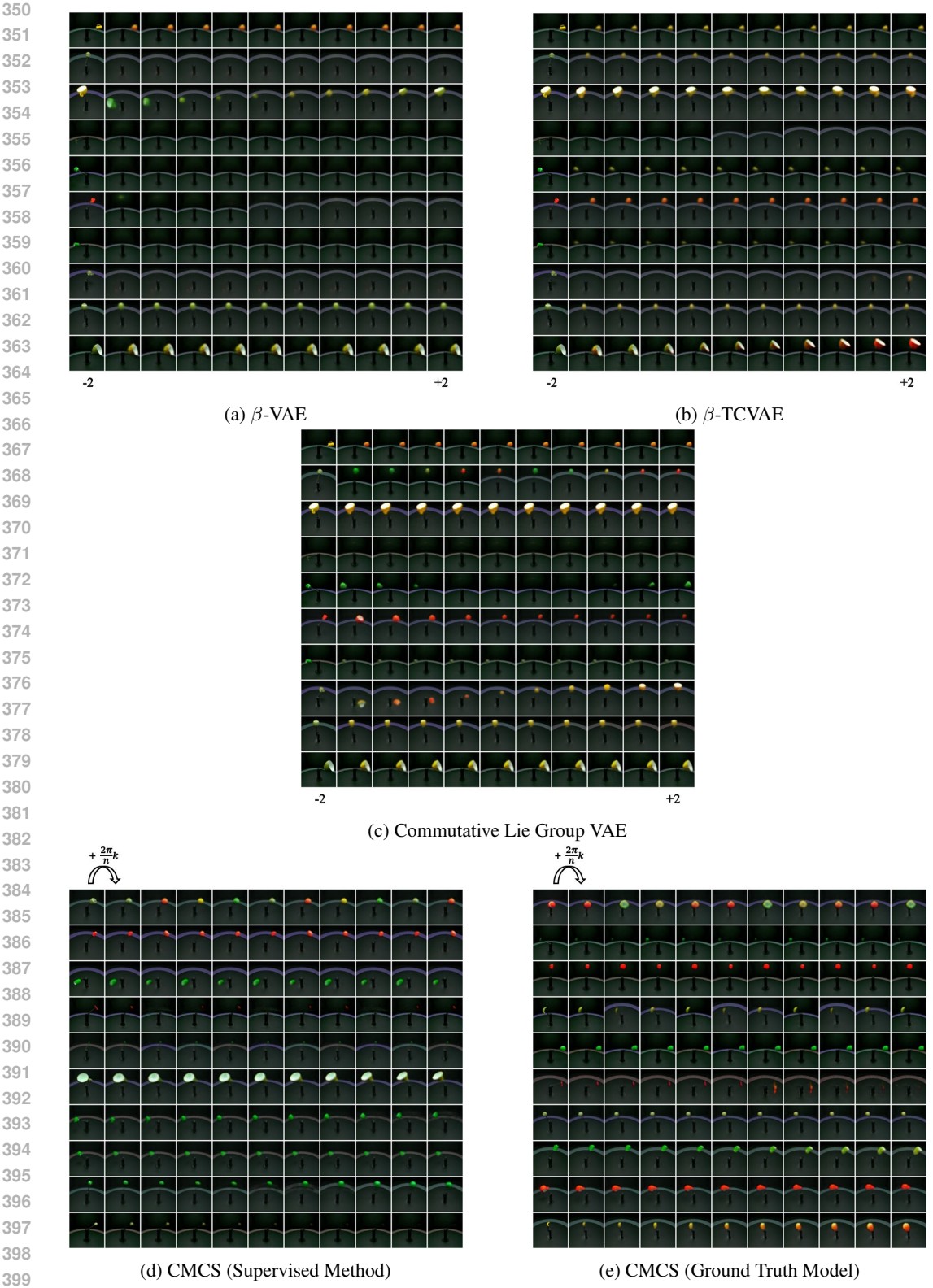

Figure 16: The $1^{st}$ column images are randomly selected from the dataset. Each row indicates each dimension of each model. $\beta$-VAE, $\beta$-TCVAE, and Commutative Lie Group VAE trace each dimension value from -2 to +2. The proposed methods apply a group action $+\frac{2\pi}{n}$ to the selected images a total of 10 times.

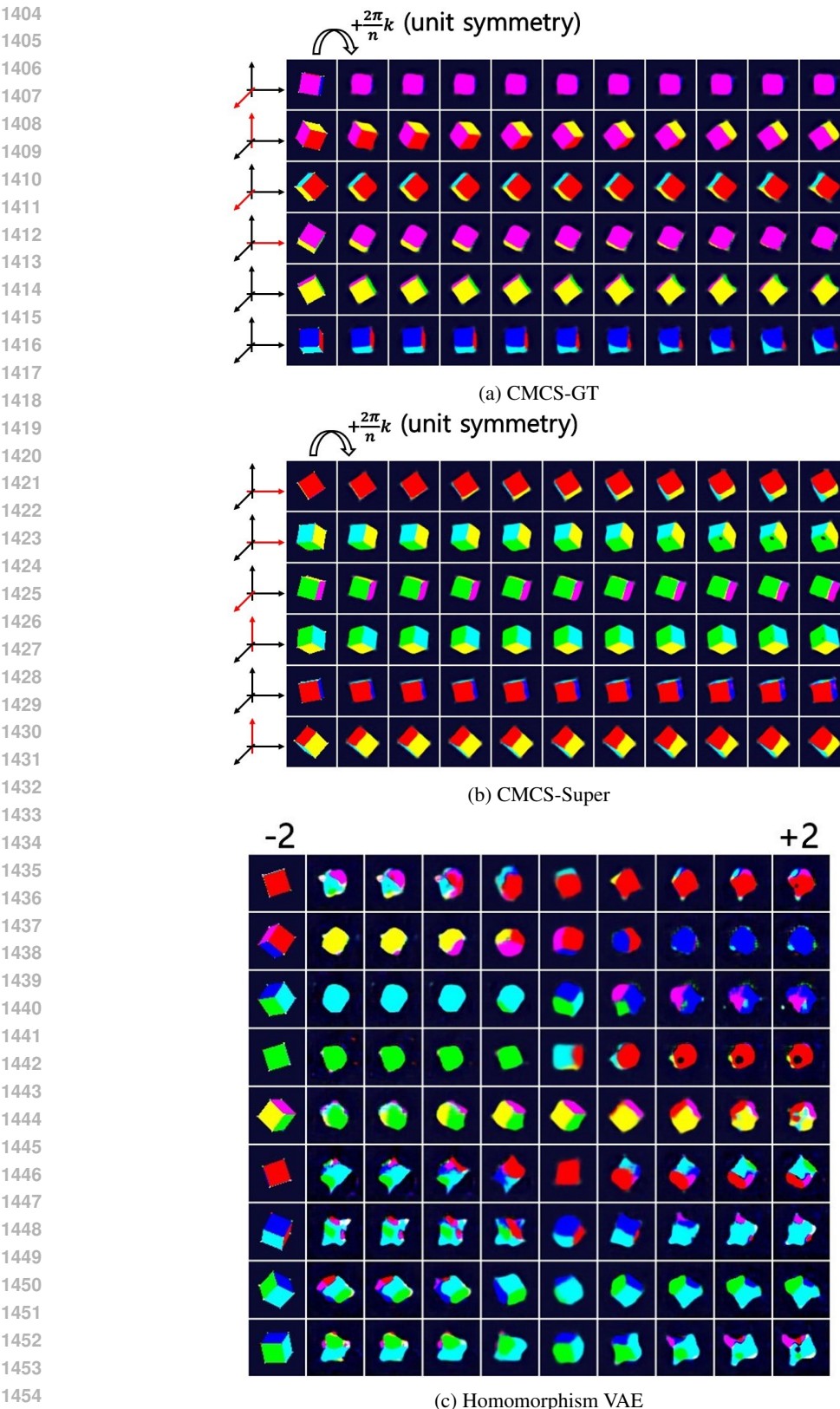

(a) CMCS-GT

(b) CMCS-Super

(c) Homomorphism VAE

Figure 17: The $1^{st}$ column images are randomly selected from the dataset. Each row indicates each dimension of each model. CMCS-GT, CMSC-Super ($\alpha$: 100.0), and homomorphism VAE trace each dimension value from -2 to +2. The proposed methods apply a group action $+\frac{2\pi}{n}$ to the selected images a total of 10 times. And red color axis is a rotation axis.

