# OpenReview forum: "Consistent Symmetry Representation over Latent Factors of Variation"
_ICLR.cc/2025/Conference — ICLR 2025 Conference Withdrawn Submission_

### Official Review · Reviewer_QDvN · 2024-10-31

**Soundness:** 1
**Presentation:** 1
**Contribution:** 1
**Rating:** 3
**Confidence:** 3

**Summary:**

The paper discusses the problem of symmetry-based representation learning. It argues that existing methods cannot obtain *consistent symmetries*, as defined in Section 2.1, and proposes the use of conformal mapping between the real space and the angle space to learn consistent symmetries.

**Strengths:**

The paper targets an interesting problem, symmetry-based representation learning. It develops a method to achieve disentanglement into single dimensions for each of the cyclic variation factors, which requires additional care in constructing a bijective mapping between the real space and the angle space.

**Weaknesses:**

The authors discuss some limitations of some previous works in disentangled representation learning in Section 2.1-2.2. However, as far as I could understand, the authors seem to have some fundamental misunderstanding about the settings and assumptions in these related works. Such misunderstanding leads to conclusions e.g. in Prop 2.1, Thm 2.2, and Thm 2.3, which do not make any sense to me. In particular,
* L107: the authors assume that the latent space can be decomposed into single dimensions $Z_i = \mathbb R$, each of which is acted on by a separate group $G_i$. This is not the setting in Higgins et al (2018), or any of the other related works mentioned in L138 (Miyato et al 2022, Marchetti et al 2023). The standard setup is that $Z = \oplus Z_i$, where each $Z_i$ can be a **multi**-dimensional vector space and is acted on by a group $G_i$. **My main objection to this paper is about this setup, and most of my following comments are related to this.** If I've misunderstood it, or if the authors think this should indeed be the correct setting to consider, please point it out before all others and I'm happy to have further discussion.
* Prop 2.1, Thm 2.2-2.3: Under the assumption in L107, it goes without saying that there is not a valid and nontrivial group action (for a cyclic group) on the latent space $Z = {\mathbb R}^{\oplus n}$. This is not a limitation of the general linear group, but rather how you have chosen to construct the latent space. Just consider the simplest case when there is only one factor of variation, $Z = \mathbb R$ and $G = SO(2)$. There is no nontrivial $G$-action on $Z=\mathbb R$, but the problem no longer exists if you allow a 2-dimensional latent space $Z=\mathbb R^2$. Also, the exponentiation of a group element ($\exp(g)$) in L145 does not make sense.
* To sum up the previous point, although Prop 2.1, Thm 2.2-2.3 might be correct, it does not offer any insight because it is based on a quite limiting assumption about the structure of the latent space, which is not used in any other related work. Also, these statements have been made in an unnecessarily complicated way with lengthy proofs, while they only reveal a self-evident fact.
* Thm 2.4: same as above - you simply cannot represent a cyclic group in $\mathbb R$. Besides, Thm 2.4 states that "if the group action is defined as $\alpha(g, z_i) = g + z_i$, which is not even a valid definition of group action of a cyclic group ($\alpha(g_1g_2, z) \not= \alpha(g_1, \alpha(g_2, z))$. I also don't find any evidence in Hwang et al (2023) that group action is defined as such vector addition.
* Cor 2.5: I don't know how to read it because symbols like $\mathcal Y$ and $\Gamma'$ are not defined nearby or in the symbol table.

Then, the method is developed based on the assumption that each single dimension in the latent space has to correspond to a factor of variation. Based on this setup, I do believe the method makes sense and addresses the limitations of this setup. However, as I've pointed out earlier, all these efforts may not be necessary since we can decompose the latent space in another way. The most straightforward solution is to replace each component $\mathbb R$ with $\mathbb R^2$ or possibly $\mathrm{SO}(2)$, which does not require constructing the conformal mapping to a complex space and mapping it back to $(\pi, \pi]$. Many other related works take a similar approach to this [1,2,3]. These works should be discussed in more detail and compared through experiments in the revised paper if possible.

The experiment results look great at first sight. However, it seems that all comparisons are still made under the limiting setup that each factor of variation can only be represented in one dimension. I feel this would limit the capability of the baseline models. It would be great to compare with some other related works that do not enforce this limitation [1,2,3].

### Other mistakes, typos & misleading points
* L120: How you describe the concept of inconsistent symmetry seems a bit confusing to me. I think for a specific task we have a fixed symmetry group $G$, and the difference is really just the different group actions on different spaces. E.g. the group acts on $F$ through an action $\alpha_F$, while it acts on $Z$ with a different action $\alpha_Z$. The current notation ($g_z', g_z''$) may suggest there are different groups and group elements, which may not be the clearest way to explain it.
* L125: Following the point above, confusion arises when you define the consistent symmetry as $g_z = \Gamma(g_F)$. But why can't this also represent the inconsistent symmetries shown in Figure 2? Concretely, $F = \\{F_0, ..., F_3\\}$, $Z = \\{Z_0, ..., Z_3\\}$, $G_F = \\{ g_F^0, g_F^1, g_F^2, g_F^3 \\} $ and $G_z = \\{ g_z^0, g_Z^1, g_Z^2, g_Z^3 \\}$. $g_F^k$ acts on $F$ by $F_i \mapsto F_{i+k}$, where $+$ is the addition modular $4$. Similarly, $g_z^k$ acts on $Z$ by $z_i \mapsto z_{i+k}$. These actions are both regular, and $\Gamma$ is simply $g_F^k \mapsto g_z^k$, which is irrelevant to how $z_i$'s are positioned on the circle in Figure 2.
* L138: GL(n) stands for general linear group. The term "general Lie group" is misleading.
* L141: "Let assume..." --> "Assume..."
* L143: The notation $GL'(n)$ does not make sense to me.
* L145: The exponential map is rarely, if ever, denoted by a bold $\mathbf e$. Use $\exp$ or simply $e^\cdot$ for matrix exponential.
* L750: $\mathfrak g$ is commonly used to refer to the Lie algebra, instead of its elements.
* Eq (8): why does the group action output a row vector? Shouldn't it be $\alpha(g, \mathbf z) = g\mathbf z$, as in Proposition A.1?
* Tab 2&3: The metrics (beta-VAE, FVM, MIG, DCI) are not explained.

### References
[1] Homomorphism Autoencoder — Learning Group Structured Representations from Observed Transitions. ICML 2023.

[2] Neural Fourier Transform: A General Approach to Equivariant Representation Learning. ICLR 2024.

[3] Latent Space Symmetry Discovery. ICML 2024.

**Questions:**

* Can you explain the differences between your experiment results and those reported in MAGANet? E.g. the results in Table 1 (dSprite), do not match those in Table 1 in the MAGANet paper.

---

> ### Author Response · Authors · 2024-11-25
> **Common Comment for Reviewer QDvN**
>
> To help reviewers clearly understand our motivation and goal, we provided the following common comments:
> 1. Motivation: An explanation of why we focus on the restricted condition of a single dimension.
> 2. Definition: A precise mathematical definition of consistent symmetry.
> ### Motivation: Dimension-wise disentangled representation
> > Disentanglement learning based on VAE has focused on **a single dimension** to contain the information of a single factor [5-11]. Group-theory-based works [12-16] have pursued the same objective, and evaluation metrics [3-6] for disentangled representations have also been developed to assess **how consistently a single factor can be represented in a single dimension**. Representing a single factor within a single dimension has notable strengths in terms of **interpretability** [1]. However, even with very simple data (sprites, 3D Shapes, etc.), this objective has yet to be fully achieved [5-16]. Thus, before addressing real-world data and general cases, **it is necessary to first discuss how to inject inductive bias in simple cases effectively**. Therefore, our work focuses on these highly restricted scenarios.
> ### Definition: Consistent Symmetry and Inconsistent Symmetry
> > Let us assume, $G_F = \\{e, g_{F_j}, g_{F_j}^2, \\ldots, g_{F_j}^k \\}$, $F_j = \\{ F_j^0, F_j^1, \\ldots, F_j^k \\}$, subset $F_j^\prime \subset  F_j$, and a function $q_\phi \circ \Omega: F \rightarrow Z$,. Then group $G_{F_j}$ acted on $F_j^\prime$ as follows: $F_j^{i+1} = g_{F_j} \circ F_j^i$. Through composite function $q_\phi \circ \Omega$, the equation $q_\phi \circ \Omega (F_j^{i+1}) = q_\phi \circ \Omega (g_{F_j} \circ F_j^i)$ translate to $z_j^{i+1} = g_{z_j}^{(i) \\rightarrow (i+1)} \circ z_j^i$. Then we define the **consistent symmetry** as $g_{z_j}^{(i) \\rightarrow (i+1)}$ are **identical regardless of $i$**, otherwise  $g_{z_j}^{(i) \\rightarrow (i+1)}$is referred to as an inconsistent symmetry.
> >
> > We revised it in ‘Definition 2.1’ in the revised version.
> ---
>
> > ### References
> >
> > [1] Yoshua Bengio. Representation learning: a review and new perspectives. TPAMI, 2013.
> >
> > [2] Xin Wnag. Disentangled Representation Learning, TPAMI 2022.
> >
> > [3] Abhishek Kumar. VARIATIONAL INFERENCEOF DISENTANGLED LATENT CONCEPTS FROM UNLABELED OBSERVATIONS. ICLR, 2018.
> >
> > [4] Cian Eastwood and Christopher K. I. Williams. A framework for the quantitative evaluation of disentangled representations. In International Conference on Learning Representations, 2018.
> >
> > [5] Irina Higgin,. beta-vae: Learning basic visual concepts with a constrained variational framework. In ICLR, 2017.
> >
> > [6] Hyunjik Kim and Andriy Mnih. Disentangling by factorising., ICML, 2018.
> >
> > [7] Diederik P Kingma and Max Welling. Auto-encoding variational bayes, 2013.
> >
> > [8] Ricky T. Q., Isolating sources of disentanglement in variational autoencoders., NeurIPS, 2018.
> >
> > [9] Huajie Shao, ControlVAE: Controllable variational autoencoder., ICML, 2020.
> >
> > [10] Yeonwoo Jeong and Hyun Oh Song. Learning discrete and continuous factors of data via alternating disentanglement., ICML, 2019.
> >
> > [11] Huajie Shao, Rethinking controllable variational autoencoders., CVPR, June 2022.
> >
> > [12] Xinqi Zhu, Commutative lie group VAE for disentanglement learning. CoRR, abs/2106.03375, 2021.
> >
> > [13] Loek Tonnaer, Quantifying and learning linear symmetry-based disentanglement., ICML, 2022.
> >
> > [14] Hee-Jun Jung, CFASL: Composite factor-aligned symmetry learning for disentanglement in variational autoencoder. TMLR, 2024.
> >
> > [15] Nikita Balabin, Disentanglement learning via topology., ICML, 2024.
> >
> > [16] Tao Yang, Groupifyvae: from group-based definition to vae-based unsupervised representation disentanglement. CoRR, abs/2102.10303, 2021.
> >
> > [17] Symmetry-based disentangled representation learning requires interaction with environments. NeurIPS 2019
> >
> > [18] Homomorphism Autoencoder — Learning Group Structured Representations from Observed Transitions. ICML 2023.
> >
> > [19] Neural Fourier Transform: A General Approach to Equivariant Representation Learning. ICLR 2024.
> >
> > [20] Latent Space Symmetry Discovery. ICML 2024.
> >
> > [21] Geonho Hwang, MAGANet: Achieving combinatorial Generalization by modeling a group action., ICML, 2023.
> >
> > [22] https://github.com/yvan/cLPR
> >
> > [23] Jaehoon Cha and Jeyan Thiyagalingam. Orthogonality-enforced latent space in autoencoders: An approach to learning
> >
> > [24] Vankov, I. I. and Bowers, J. S. Training neural networks to encode symbols enables combinatorial generalization. Philosophical Transactions of the Royal Society B, 375 (1791):20190309, 2020.
> >
> > [25] Irina Higgins, Towards a definition of disentangled representations. CoRR, abs/1812.02230, 2018.
> >
> > [26] Yingheng Wang, InfoDiffusion: Representation Learning Using Information Maximizing Diffusion Models, ICMLR, 2023.
> >
> > [27] Tero Karras, A Style-Based Generator Architecture for Generative Adversarial Networks, CVPR, 2019.

---

> ### Author Response · Authors · 2024-11-25
> **Response to Reviewer QDvN # 1**
>
> R4 W1.
> > L107: the authors assume that the latent space can be decomposed into single dimensions $Z_i=\mathbb{R}$, each of which is acted on by a separate group $G_i$. This is not the setting in Higgins et al (2018), or any of the other related works mentioned in L138 (Miyato et al 2022, Marchetti et al 2023). The standard setup is that $Z=⊕Z_i$, where each $Z_i$ can be a **multi**-dimensional vector space and is acted on by a group $G_i$. **My main objection to this paper is about this setup, and most of my following comments are related to this**. If I've misunderstood it, or if the authors think this should indeed be the correct setting to consider, please point it out before all others and I'm happy to have further discussion. We have revised the parts of our paper that may have caused misunderstanding.
>
> R4 A1.
> > We acknowledge that the mentioned $Z_i$ is defined in a multi-dimensional form. However, we focus on the **dimension-wise disentangled representation** as discussed in [2]. Many previous works [5-16] have also focused on this form of disentangled representation. Furthermore, since the metrics for evaluating disentangled representation assess how consistently a single factor is encapsulated within a single dimension [3-6], we considered the highly restricted case of $Z_i= \mathbb{R}$, which corresponds to **dimension-wise disentangled representation**. We have revised the parts of our paper that may have caused misunderstanding and we highlighted them on Lines 106-107 in the revised version.
>
> ---
>
> R4 W2.
> > Prop 2.1, Thm 2.2-2.3: Under the assumption in L107, it goes without saying that there is not a valid and nontrivial group action (for a cyclic group) on the latent space $Z=\mathbb{R}^{⊕n}$. This is not a limitation of the general linear group, but rather how you have chosen to construct the latent space. Just consider the simplest case when there is only one factor of variation, $Z=\mathbb{R}$ and $G=SO(2)$. There is no nontrivial G-action on $Z=\mathbb{R}$, but the problem no longer exists if you allow a 2-dimensional latent space $Z=\mathbb{R}^2$. Also, the exponentiation of a group element $(exp⁡(g))$ in L145 does not make sense.
>
> R4. A2.
> > The **essential point** of mapping to circular latent spaces is to **achieve a cyclic structure in a single dimension** rather than in the multi dimensions. As mentioned in **‘Common Comments for Reviewer QDvN’**, we are considering the highly restricted case where $z=\mathbb{R}$. It means that our goal is to identify an arbitrary group $G$ that acts on $z=\mathbb{R}$ such that it **satisfies $g_z \circ \infty = -\infty$**. To implement this arbitrary group $G$, we map a single dimension to circular latent spaces through a bijective function for an **isomorphism** to preserve the symmetry structure of a single dimension ($f(g⋅z)=so(2)⋅f(z)$.)
> > In the case you mentioned, where $Z= \mathbb{R}^2$ and $G=SO(2)$, applying the group action results in the alteration of values in **two dimensions**, which falls outside the scope of our approach (dimension-wise disentangled representations).
> >
> > Additionally, **the group element $g$ considered in Line 145 is not all the case of $GL(n)$**, but, as mentioned in Proposition 2.1, it is an element of $GL(n)$ that **affects only a single dimension of the latent vector $z$**. While it is trivial that Line 147 would not satisfy the equation in the absence of all the conditions we proposed, under the specified conditions, Line 147 is satisfied. This is demonstrated through Eq. (9) in Appendix A.1 (Lines 791-807).
>
> ---
>
> R4 W3.
>
> > To sum up the previous point, although Prop 2.1, Thm 2.2-2.3 might be correct, it does not offer any insight because it is based on a quite limiting assumption about the structure of the latent space, **1)** which is not used in any other related work. Also, **2)** these statements have been made in an unnecessarily complicated way with lengthy proofs, while they only reveal a self-evident fact.
>
> R4 A3.
> > 1) **Previous works** directly related to our research have all focused on implementing **dimension-wise disentangled representation [5-11]**. Group-theory-based works [12-16] have **also pursued the same types of disentangled representation** [2] as ours.
> 2) Our objective has been limited to the highly restricted case, dimension-wise disentangled representation. In previous works that introduced symmetry as an inductive bias to achieve the dimension-wise disentangled representation, **the appropriate symmetry was applied without sufficient discussion**. Therefore, our contribution lies in providing a guideline **for identifying suitable symmetries** for dimension-wise disentangled representation.

---

> ### Author Response · Authors · 2024-11-25
> **Response to Reviewer QDvN # 2**
>
> R4 W4.
> > **1)** Thm 2.4: same as above - you simply cannot represent a cyclic group in R. Besides, **2)** Thm 2.4 states that "if the group action is defined as $\alpha(g, z_i)=g+z_i$, which is not even a valid definition of group action of a cyclic group $(\alpha(g_1g_2,z) \neq \alpha(g_1,\alpha(g_2,z))$. **3)** I also don't find any evidence in Hwang et al (2023) that group action is defined as such vector addition.
>
> R4 A4.
> > 1.  As mentioned R4 A2., our goal is to find an arbitrary group $G$ that can enable $Z$ to have a cyclic structure when $Z = \mathbb{R}$, and we have implemented this using conformal mapping.
> 2.  It appears there was a misunderstanding due to our insufficient definition of the group. We define $g_i$ as $g_i \in \mathbb{R}^n$, where the group $g_i$ acts on the latent vector $z \in \mathbb{R}^n$ such that $\alpha(g_i, z) = g_i + z$. The composition of two elements is defined as $g_i + g_j$. Therefore, the equation $\alpha(g_1 g_2, z) = \alpha(g_1, \alpha(g_2, z)) = g_1 + g_2 + z$, as mentioned, is satisfied.
> 3.  We apologize for providing an incorrect reference. The method described as "latent addition" is discussed in [15]. In this study, symmetry is similarly defined as **affecting only each dimension**, and the proposed group action can also be expressed through vector addition. However, the method presented in [15] **cannot perform the transformation $\infty \to -\infty$**. It is trivial that defining vector addition makes it impossible to implement a cyclic group, so we have revised the theorem to a corollary.
>
> ---
>
> R4 W5.
>
> > Cor 2.5: I don't know how to read it because symbols like $Y$ and $\Gamma^\prime$ are not defined nearby or in the symbol table.
>
> R4 A5.
> > To avoid misleading, we left the defined notation.
> > $\mathcal{Y}$ is a another **vector space**, and $\Gamma^\prime$ is an **endomorphism** that $\Gamma^\prime$: $G_Z \rightarrow G_y$. And we add these notations in Table 4 in Appendix A.
>
>
> R4. W6.
>
> > Then, the method is developed based on the assumption that each single dimension in the latent space has to correspond to a factor of variation. Based on this setup, I do believe the method makes sense and addresses the limitations of this setup. However, as I've pointed out earlier, all these efforts may not be necessary since we can decompose the latent space in another way. The most straightforward solution is to replace each component R with R2 or possibly SO(2), which does not require constructing the conformal mapping to a complex space and mapping it back to (π,π]. Many other related works take a similar approach to this [18-20]. These works should be discussed in more detail and compared through experiments in the revised paper if possible.
> >
> > The experiment results look great at first sight. However, it seems that all comparisons are still made under the limiting setup that each factor of variation can only be represented in one dimension. I feel this would limit the capability of the baseline models. It would be great to compare with some other related works that do not enforce this limitation [18-20].
> >
>
> R4 A6.
>
> > For the reviewer’s requirements, we reproduce the ‘Homomorphism VAE [18] methods and evaluate the beta-VAE, FVM, MIG, and DCI metrics on the dSprites dataset with ten seeds in the revised version Table 6. (page 23). Also, we extend the dataset to evaluate the SO(3) case, the results are presented in Fig.17 (page 27 in the revised version).
> >
> > We add the ‘Homomorphism VAE [18] and Groupified VAE [16], which utilizes the same function to implement unit circle as in [17], methods for reviewer’s requirements and **evaluate the beta-VAE, FVM, MIG, and DCI metrics** on the dSprites dataset using ten seeds as shown in Table 6 of the revised version. (page 23 in the revised version).
> >
> >\\begin{array}{c|c|c|c|c}
> \\hline
> dSprites & beta-VAE & FVM & MIG & DCI \\\\
> \\hline
> Homomorphism VAE [18] & 18.80(\\pm 5.75) & 30.24(\\pm 12.18) & 0.39 (\\pm 0.76) & 1.35 (\\pm 1.12) \\\\
> Groupified VAE [16] & 79.30(\\pm 9.23) & 69.75(\\pm 13.66) & 21.03(\\pm 9.20) & 31.08(\\pm 10.87) \\\\
> CMCS-SP & \\textbf{91.40}(\\pm 4.99) & \\textbf{93.74}(\\pm 1.82) & \\textbf{51.02}(\\pm 2.42) & \\textbf{64.69} (\\pm 1.55) \\\\
> \\hline
> \\end{array}
> >
> > As expected, our model is **much better** than Homomorphism VAE in qualitative results (Table 6 on page 23 in the revised paper), and also our model contains each axis rotation **in a single dimension** (Fig. 17 on page 27 in the revised paper).
>
> ---

---

> ### Author Response · Authors · 2024-11-25
> **Response to Reviewer QDvN # 3**
>
> > L120: How you describe the concept of inconsistent symmetry seems a bit confusing to me. I think for a specific task we have a fixed symmetry group $G$, and the difference is really just the different group actions on different spaces. E.g. the group acts on $F$ through an action $\alpha_F$, while it acts on $Z$ with a different action $\alpha_Z$. The current notation $(g_z^\prime, g_z^{\prime \prime})$ may suggest there are different groups and group elements, which may not be the clearest way to explain it.
>
> R4 A7.
> > We define the **consistent symmetry** mathematically in the **Common Comment for All Reviewers** to avoid repetition.  Also, [21] **demonstrates the existence of inconsistent symmetries** in section 5.6.
> Therefore, we focus on the **impact of given consistent symmetry** on disentanglement and combinatorial generalization performance, addressing the inconsistency issue in the previous method through unsupervised learning [12, 14, 21].
>
> ---
>
> R4 W8.
>
> > L125: Following the point above, confusion arises when you define the consistent symmetry as $g_z=\Gamma(g_F)$. But why can't this also represent the inconsistent symmetries shown in Figure 2? Concretely, $F=\\{F_0,...,F_3\\}, Z=\\{Z_0,...,Z_3\\}$, $G_F=\\{g_F^0,g_F^1,g_F^2,g_F^3\\}$ and $G_z=\\{g_z^0,g_z^1,g_z^2,g_z^3\\}$. $g_F^k$ acts on $F$ by $F_i \rightarrow F_{i+k}$, where $+$ is the addition modular 4. Similarly, $g_z^k$ acts on Z by $z_i \rightarrow z_{i+k}$. These actions are both regular, and $\Gamma$ is simply $g_F^k \rightarrow g_z^k$, which is irrelevant to how $z_i$'s are positioned on the circle in Figure 2.
>
> R4 A8.
> > Let us explain why it can't represent the consistent symmetry on your explanation.
> > First, we define each components as $F, Z, G_F,$ and $G_z$ are consist of $F = F_1 \times F_2 \times, … \times F_n, Z = Z_1 \times Z_2 \times, … \times Z_n, G_F = G_{F_1} \times G_{F_2} \times, … \times G_{F_n}$, and $G_z = G_{z_1} \times G_{z_2} \times, … \times G_{z_n}$ respectively. And each decomposed set $F_j, Z_j, G_{F_j},$ and $G_{z_j}$ are consist of $F_j = \\{F_j^0, F_j^1, …, F_j^2\\}, Z_j = \\{Z_j^0, Z_j^1, …, Z_j^2\\}, G_{F_j} = \\{e, g_{F_j}, g_{F_j}^2, … g_{F_j}^k\\},$ and $G_{z_j} = \\{e, g_{z_j}, g_{z_j}^2, …, g{z_i}^k\\}$.
> Therefore, it is impossible that changes $F_j \rightarrow F_{j+k}$ instead changes exist from $F_j^0 \rightarrow F_j^i$ as $G_{F_j}$ is acted. As we defined each component $F, Z, G_F,$ and $G_z$, then there is no inconsistent symmetries.
>
> ---
>
>
> R4 W9.
> >* L138: GL(n) stands for general linear group. The term "general Lie group" is misleading.
> >* L141: "Let assume..." --> "Assume..."
> >* L145: The exponential map is rarely, if ever, denoted by a bold e. Use exp or simply e⋅ for matrix exponential.
> >* L750: g is commonly used to refer to the Lie algebra, instead of its elements.
> >* Eq (8): why does the group action output a row vector? Shouldn't it be α(g,z)=gz, as in Proposition A.1?
>
> R4 A9.
> > We revised all comments in the revised paper.
>
> ---
>
> R4 W 10.
>
> >* L143: The notation GL′(n) does not make sense to me.
>
> R4 A10.
>
> > We apologize for the misrepresentation that caused misunderstanding. We define the $GL^\prime(n)$ as $GL^\prime(n) = \\{ e^M | M \in \mathbb{R}^{n\times n}\\}$ (we highlight it on L146 in the revised version).
>
> ---
>
> R4 W 11.
>
> >* Tab 2&3: The metrics (beta-VAE, FVM, MIG, DCI) are not explained.
>
> R4 A11.
>
> > These four metrics are mainly used to evaluate the dimension-wise disentanglement representations, we put the references of each metric in Appendix B.4 (we highlight it on L1046-1047 in the revised version).
> ---
>
> R4 Q1.
> > * Can you explain the differences between your experiment results and those reported in MAGANet? E.g. the results in Table 1 (dSprite), do not match those in Table 1 in the MAGANet paper.
>
> R4 QA1.
> > We obtained the results of MAGANet approach by replacing its Glow model part to VAE model for ensuring consistency of model structures between baselines and our approaches. Additionally, the evaluation metric for the 3DShapes and MPI3D dataset is **MSE loss** instead of BCE loss.

---

> ### Comment · Area_Chair_U95m · 2024-11-27
> **Rebuttal Response**
>
> Dear Reviewer,
> Do you mind letting the authors know if their rebuttal has addressed your concerns and questions? Thanks!
> -AC

---

> ### Comment · Reviewer_QDvN · 2024-11-27
> **Response to authors**
>
> I thank the authors for the detailed response and appreciate their efforts to provide additional experiment results.
>
> Unfortunately, I still find it hard to agree with the core argument of this paper that disentangled representation means dimension-wise group action. While I acknowledge that some papers discuss this setting [2], I believe many works on symmetry-based representation learning do not place this restriction, including the seminal work [25]. Also, some of the works that the authors mentioned to ``focus on dimension-wise representation'' actually do not. I have not read through all the references, but I will take Commutative Lie Group VAE [12] which is more familiar to me as an example. They have explicitly mentioned that the representation is more than one dimensional, e.g. in Eqn (4) where the Lie algebra has a matrix representation. So I would encourage the authors to be more careful citing other papers and avoid making faulty statements about other people's works.
>
> I also read the revised definition of consistent and inconsistent symmetry but still find it unclear. I think it is important to distinguish between the group itself, which is an algebraic structure and assumed to be a cyclic group in this paper, and its group actions on different spaces. Currently, it seems that the action of $g_{F_j}$ on the latent space $Z_j$ is expressed as $g_{z_j}$, which is very confusing because it looks like another element of another group.
>
> There are more comments I could think of regarding other revisions such as Sec 2.2 and the additional experiments. However, I think the above are the most critical points that the authors need to prioritize, so I will end my response here. I hope the authors can consider my points and improve their paper throughout. As it stands, I will maintain my original score.

---

> > ### Author Response · Authors · 2024-11-28
> > **Response to Reviewer QDvN # 4**
> >
> > Reviewer’s Comment 1
> > > Unfortunately, I still find it hard to agree with the core argument of this paper that disentangled representation means dimension-wise group action.
> >
> >
> > Author’s Answer 1
> > >  Thank you for your prompt response. However, our point has not been fully conveyed, so we would like to offer further clarification:
> > >
> > > Our core argument is NOT that “disentangled representation means dimension-wise group action.” The concept of disentangled representation varies significantly depending on the research area. Our focus is on advancing research that specifically explores learning dimension-wise group actions for disentanglement [12-17] and its interpretation . We do not intend to claim that our method addresses the unrestricted setting.
> > >
> > > In dimension-wise disentangled representation [2], each dimension contains a single latent factor. If the group action is not dimension-wise, then there is no group action that affects only a single factor. Also, if the group action affects a single factor variation through the multi-dimension of the latent vector, then it also contradicts the dimension-wise disentangled representation. This is why we focus on dimension-wise group action alongside dimension-wise disentangled representation.
> > >
> > > We hope that this distinction in research areas is given more serious consideration in your evaluation,
> >
> > ---
> >
> > Reviewer’s Comment 2
> > > While I acknowledge that some papers discuss this setting [2], I believe many works on symmetry-based representation learning do not place this restriction, including the seminal work [25]. Also, some of the works that the authors mentioned to ``focus on dimension-wise representation'' actually do not. I have not read through all the references, but I will take Commutative Lie Group VAE [12] which is more familiar to me as an example. They have explicitly mentioned that the representation is more than one dimensional, e.g. in Eqn (4) where the Lie algebra has a matrix representation. So I would encourage the authors to be more careful citing other papers and avoid making faulty statements about other people's works.
> >
> > Author’s Answer 2
> > > The mentioned Commutaitve Lie Group VAE [12] can be regarded as dimension-wise diesentangled representation setting. Becuase Commutative Lie Group VAE [12] optimizes the Hessian matrix to be Idendity matrix as shown in equation (7): $\frac{\mathrm{d}^2 f(z)}{\mathrm{d} z_i \mathrm{d} z_j} = 0$. This implies that each dimension is independent. Moreover, **[12] explicitly states, "since the variation controlled by a dimension should not be a function of another dimension (independent)"** [12] in the paper. Therefore, Commutative Lie Group VAE [12] also considers a dimension-wise disentangled representation. Furthermore, this model evaluates the FVM, SAP, MIG, and DCI metrics, which represent the quality of a dimension-wise disentangled representation.
> >
> > > We agree with the point made in [25] and other referenced papers that there are no strict limitations imposed on disentangled representation. However, the fact that many papers have not considered dimension-wise disentangled representation does not render our research unnecessary. As we have already indicated in references [5-17], these papers do take dimension-wise disentangled representation into account. Within this context, our main contribution lies in theoretically addressing and applying an appropriate dimension-wise group action to the groups published so far, thereby improving existing performance. Moreover, there has been very little prior research addressing disentanglement learning and combinatorial generalization simultaneously. Our work is the first to improve both tasks through a single approach, which is another key contribution of our study.
> >
> > ---
> >
> > Reviewer’s Comment 3
> > > I also read the revised definition of consistent and inconsistent symmetry but still find it unclear. I think it is important to distinguish between the group itself, which is an algebraic structure and assumed to be a cyclic group in this paper, and its group actions on different spaces. Currently, it seems that the action of gFj on the latent space Zj is expressed as gzj, which is very confusing because it looks like another element of another group.
> >
> > Author’s Answer 3
> >  > $g_{F_j}$ and $g_{z_j}$ are different group elements. $g_{F_j}$ acts on the latent factors $F_j$, and $g_{z_j}$ acts on the latent vectors $z_j$. As described in the **'Cyclic Semantics of Dataset Sapce'** paragraph in Section 2.1, we defined each space and the cyclic group $G_z$ in the **'Disentangled Representation on Latent Vector Space'** paragraph in Section 2.1.

---

> > > ### Comment · Reviewer_QDvN · 2024-11-28
> > >
> > > I still don't agree that Commutative Lie Group VAE [12] implements dimension-wise representation. The Hessian penalty is imposed on the group representation, i.e. Eqn (8), to encourage the commutative property. While a dimension-wise representation implies the group is commutative, the reverse is not true. In the authors' quote to the original paper, the dimensions refer to those corresponding to the basis vectors of the Lie algebra, not those of the vector space acted on by the group. It can also be found in Table 2 in [12] that they used matrix representations.

---

> > > > ### Author Response · Authors · 2024-11-29
> > > > **Response to Reviewer QDvN # 5**
> > > >
> > > > > Before addressing your comments, we kindly ask you to consider the following points:
> > > > >
> > > > > 1. The core focus of this paper is not on evaluating the accuracy of the references cited, but rather on demonstrating how effectively latent factors are represented under the specific, limited conditions explicitly addressed in the study. While we fully agree that it is important to assess the appropriateness of the motivation, if there happens to be an error in our citation of a particular reference or a misunderstanding of its content, we believe this can be rectified through revisions.
> > > > >
> > > > > 2. Moreover, **the main contribution** of this paper lies in presenting a guideline that simultaneously addresses disentanglement learning and combinatorial generalization tasks—**two tasks that are rarely tackled together**. We are concerned that the discussion has not adequately engaged with this contribution and instead seems to focus on the critique that existing methods do not deal with "restricted conditions". This raises a question about whether the discussion appropriately addresses our work's key contributions.
> > > > >
> > > > > 3. Research on dimension-wise disentangled representation is ongoing, and we believe we have provided sufficient references to support this. Additionally, if it is still perceived as a simple matter of increasing the number of dimensions, we kindly ask you to review the extended experiments we have provided. We have already addressed this in **"R4 A6"**.
> > > >
> > > >
> > > > ---
> > > >
> > > >
> > > > Reviewer’s Comment
> > > > >  I still don't agree that Commutative Lie Group VAE [12] implements dimension-wise representation. The Hessian penalty is imposed on the group representation, i.e. Eqn (8), to encourage the commutative property. While a dimension-wise representation implies the group is commutative, the reverse is not true. In the authors' quote to the original paper, the dimensions refer to those corresponding to the basis vectors of the Lie algebra, not those of the vector space acted on by the group. It can also be found in Table 2 in [12] that they used matrix representations.
> > > >
> > > >
> > > >
> > > >
> > > > Author’s Answer
> > > > > We hope the reviewer to check the following evidences supporting that CLG is based on dimension-wise settings.
> > > > >
> > > > > 1. Eq. 8 encourages not only the commutative property but also Eq. 7. As [12] explicitly states “ **Our Lie algebra parameterization is compatible with this constraint** and we have the following proposition: Eq. 8.”[12] on page 4, right below Eq. 7. Therefore, the Eq. 8 is optimized for Eq. 7.
> > > > > 2. [12] explicitly states, “It can be expected that the larger the group representation is, **the more likely the subgroups (indexed by coordinates $t_i$'s)** can control different variations (due to the increased sparsity in a larger space)” in the “How the Group Representation Size Affects Disentanglement” paragraph on pages 6-7. This implies that $t$ is a dimension-wise disentangled representation because a single dimension $t_i$ controls different variations. Also, the Disentangled representation is not only defined on the latent vector but also in other spaces. Also, the result in Table 2 is evaluated using the latent variable $t$. These metrics always use vectors, not matrices [3-6]. As shown in Fig. 5, the latent traversals are controlled by each $t_i$, where the dimension of $t$ is 10 not 100, and it is equal to the number of rows of Fig. 5.
> > > > >
> > > > > Beyond this specific method, we would like to point out that numerous studies have explored disentanglement approaches within dimension-wise settings.

---

> > ### Author Response · Authors · 2024-11-28
> > **Response to Reviewer QDvN # 5**
> >
> > Reviewer’s Comment 4
> > > There are more comments I could think of regarding other revisions such as Sec 2.2 and the additional experiments. However, I think the above are the most critical points that the authors need to prioritize, so I will end my response here. I hope the authors can consider my points and improve their paper throughout. As it stands, I will maintain my original score.
> >
> >
> >
> >
> > Author’s Answer 4
> > > Thank you for engaging in this lengthy discussion and for your thoughtful consideration. However, we still feel that the contributions to disentanglement and combinatorial generalization have not been adequately evaluated within the appropriate research context, as they seem to have been overshadowed by the reviewers' focus on presenting a more generalized theorem. Regardless of the final decision, we hope that this point will be conveyed to the reviewers.

---

### Official Review · Reviewer_ZKWo · 2024-11-01

**Soundness:** 3
**Presentation:** 3
**Contribution:** 3
**Rating:** 6
**Confidence:** 3

**Summary:**

This paper addresses the challenge of achieving consistent symmetry representation, where identical semantic transformations are represented consistently via an isomorphism $\Gamma: G_F \to G_z$, where $G_F$ denotes the group of cyclic semantic transformations and $G_z$ represents its corresponding representation. The authors argue that conventional symmetry-aware representation methods, such as matrix Lie groups, vector addition, or surjective mappings, do not ensure the consistent symmetry representation. To address this, they introduce Conformal Mapping for Consistent Symmetries (CMCS), a method that maps latent variables to the complex plane using conformal mappings, thereby establishing a cyclic group action on a fixed grid (a codebook for symmetries). The proposed model operates under three scenarios: ground truth (where the symmetry information is accessible), supervised, and semi-supervised (where symmetry information is extracted from labels). Experimental validation on standard benchmarks like dSprites and 3D Shapes demonstrates that CMCS outperforms baseline models in both combinatorial generation and disentanglement learning tasks.

**Strengths:**

The paper is well-motivated and interesting. It demonstrates that the inductive biases frequently used in previous symmetry-aware representations are insufficient to satisfy the consistency symmetry condition, and proposes how to efficiently improve them.
***
The proposed method, CMCS, appears novel. I find the use of conformal mapping to represent real space latent variables in the complex plane and define group action through this approach to be sufficiently novel and technically sound. The conformal mapping-based representation suggested by the authors seems promising, with various potential applications, and I would like to rate it highly for novelty. However, as I have not exhaustively reviewed related literature, this evaluation may be subject to adjustment.
***
The experimental results of this paper leave me with mixed feelings. The main methodology of this paper is based on ground truth-based or supervised/semi-supervised learning, yet it is primarily compared with unsupervised learning methods, which gives a somewhat unfair impression (e.g., Table 2). However, the paper’s main claims are effectively demonstrated experimentally, specifically (1) that existing approaches for handling symmetry (e.g., GL(n)) are not sufficiently efficient (e.g., Table 3) and (2) that the proposed CMCS can go beyond the Pareto front in the reconstruction-dismantlement trade-off (e.g., Figure 7).

**Weaknesses:**

The proposed method relies heavily on guidance (ground truth or labels) for learning symmetry, raising questions about the fairness of comparing it with other unsupervised learning-based approaches. Although the authors propose a semi-supervised method, they still assume that 50% of the data is labeled, significantly higher than the label proportion typically used in most semi-supervised and unsupervised learning studies.

**Questions:**

Could the authors evaluate the performance variation of CMCS-Semi by adjusting the accessible label ratio $p$? I believe demonstrating robust performance even at a low $p$ would strengthen the authors' claims significantly.

---

> ### Author Response · Authors · 2024-11-25
> **Response to Reviewer ZKWo**
>
> R3 Q1.
> > Could the authors evaluate the performance variation of CMCS-Semi by adjusting the accessible label ratio p? I believe demonstrating robust performance even at a low p would strengthen the authors' claims significantly.
>
> R3 A1.
> > To provide the results asked by this comment, we added experiment that the impact of the label ratio $p \in \\{0.1, 0.2, 0.4 \\}$ on dSprites and 3DShapes datasets. As shown in the table, most cases still outperform baseline (Ada-GVAE). Additionally, the results are robust to changes in ratio except for the DCI metric.
> >
> > We update the results on page 24 in the revised paper.
> >\\begin{array}{c|c|c|c|c|c|c|c|c}
> \\hline
> dSprites& beta-VAE & FVM & MIG & DCI \\\\
> \\hline
> Ada-GVAE & 83.60(\\pm 2.61) & 83.67(\\pm 2.97) & 21.34(\\pm 5.35) & 47.26(\\pm 6.50) \\\\
> \\hline
> CMCS-semi (0.1) & 88.60(\\pm 7.72) & 83.36(\\pm 3.51) & 14.71(\\pm 1.25) & 23.46(\\pm 1.69) \\\\
> CMCS-semi (0.2) & 89.78(\\pm 6.67) & 83.88(\\pm 2.76) & 23.03(\\pm 2.25) & 28.07(\\pm 1.40) \\\\
> CMCS-semi (0.4) & 88.20(\\pm 5.92) & 83.49(\\pm 2.22) & 31.87(\\pm 2.02) & 39.44(\\pm 0.79) \\\\
> CMCS-semi (0.5) & 87.00(\\pm 7.07) & 84.50(\\pm 1.41) & 31.95(\\pm 2.40) & 39.36(\\pm 1.49) \\\\
> \\hline
> \\end{array}
> >
> >\\begin{array}{c|c|c|c|c|c|c|c|c}
> \\hline
> 3DShapes& beta-VAE & FVM & MIG & DCI \\\\
> \\hline
> Ada-GVAE & 72.75(\\pm 6.50) & 59.81(\\pm 6.14) & 24.77(\\pm 7.48) & 64.57(\\pm 4.04) \\\\
> \\hline
> CMCS-semi (0.1) & 86.80(\\pm 3.90) & 84.05(\\pm 2.66) &55.17(\\pm 2.18) & 61.79(\\pm 3.15) \\\\
> CMCS-semi (0.2) & 85.00(\\pm 13.11) & 83.00(\\pm 7.29) & 54.20(\\pm 13.35) & 60.26(\\pm 12.00) \\\\
> CMCS-semi (0.4) & 86.40(\\pm 6.98) & 87.61(\\pm 7.09) & 61.47(\\pm 9.51) & 67.87(\\pm 8.39) \\\\
> CMCS-semi (0.5) & 95.00(\\pm 7.07) & 88.81(\\pm 1.41) & 57.94(\\pm 16.52) & 72.14(\\pm 3.23) \\\\
> \\hline
> \\end{array}

---

> ### Comment · Area_Chair_U95m · 2024-11-27
> **Rebuttal Response**
>
> Dear Reviewer,
> Do you mind letting the authors know if their rebuttal has addressed your concerns and questions? Thanks!
> -AC

---

### Official Review · Reviewer_QFQU · 2024-11-04

**Soundness:** 2
**Presentation:** 3
**Contribution:** 2
**Rating:** 6
**Confidence:** 3

**Summary:**

The paper proposes a method for learning disentangled representations in variational autoencoders by enforcing consistent symmetry representations through conformal mapping to circular latent spaces. The authors identify limitations in existing approaches using matrix exponentials, vector addition, and surjective mappings, and propose a solution using the Cayley transform to map latent vectors to the unit circle. The method is evaluated on three datasets (dSprites, 3D Shapes, and MPI3D) and shows improvements in both disentanglement metrics and combinatorial generalization tasks.

**Strengths:**

The paper is generally well-written with clear figures, and makes a good connection between theoretical framework and experimental validation.

- Identifying the importance of consistency in how transformations are represented
- Attempting to bridge disentanglement and combinatorial generalization through symmetry
- Provides a concrete solution for maintaining cyclic behavior in latent spaces
- The Cayley transform implementation, while overcomplicated, is mathematically valid for their restricted setting
- The experiments have comprehensive evaluation using multiple metrics
- Clear demonstration of improvements in their specific setting
- Good ablation studies showing the impact of different components

**Weaknesses:**

The theoretical framework is limited to independent cyclic symmetries, and the mathematical treatment contains several confusing or incorrect formulations. While the paper shows promising results for simple cyclic transformations, it carefully avoids more challenging cases like coupled symmetries or full 3D rotations, where the proposed method would face fundamental topological limitations.
If the paper would admit to these restrictive simplifications, it could be presented as a first step for cases where single latent dimensions represent disentangled reps.
In detail, my issues are:

1. Theoretical Framework:
- The mathematical treatment of Lie groups and their actions is confused, particularly in their handling of $GL'(n)$ and matrix exponentials
- The fundamental assumption that symmetries must act on single dimensions is too restrictive and ignores basic topological constraints (e.g., impossibility of faithfully representing $SO(3)$, i.e. symmetries of 3D objects, through independent $SO(2)$ actions, i.e. the largest compact group that can act on a single latent dim.)
- The "conformal mapping" solution is unnecessarily complicated for what is essentially just mapping to circular latent spaces

2. Limited Scope and Misleading Presentation:
- The method only handles independent cyclic symmetries, but this crucial limitation isn't clearly acknowledged
- Dataset choice carefully avoids challenging cases (like full 3D rotations) that would reveal the method's limitations
- Presents as a general solution for symmetries in latent spaces when it's actually quite restricted

3. Novelty and Positioning:
- Insufficient discussion of prior work on group actions in latent spaces
- Doesn't clearly position their contribution relative to existing symmetry-based approaches
- The core idea of using group actions for consistent representations isn't novel (Caselles-Dupré et al "Symmetry-based disentangled representation learning requires interaction with environments." NeurIPS 2019), though their specific implementation for cyclic groups is

4. Technical Issues:
- Several confusing or incorrect mathematical formulations, particularly in Section 2 (see below)
- The proofs about limitations of existing methods are flawed due to overly restrictive assumptions
- The "consistent symmetry" definition could be much more simply stated

5. Experimental Validation:
- Results are only shown for cases that fit their restricted framework
- Missing experiments with more challenging symmetries (e.g. more general 3D rotations).
- No comparison with methods specifically designed for handling group actions

### Technical Issues details

1. Mathematical Formulation Problems:
- Proposition 2.1 and related theorems contain serious mathematical confusions:
  * Mix up elements of Lie groups with elements of Lie algebras
  * Write nonsensical equations like $e^g z = eIgz + v'$ where exponentials are applied inconsistently
  * Attempt to exponentiate group elements rather than Lie algebra elements
  * Fail to properly distinguish between matrix exponential and scalar exponential

2. Flawed Analysis of Matrix Groups:
- Their critique of using $GL(n)$ ignores well-known solutions in Lie theory:
  * Don't consider proper compact subgroups like $O(n)$ where cyclic behavior is natural
  * Miss the fact that elements of $\mathfrak{so}(n)$ can easily generate cyclic groups via exponential map
  * Make unnecessarily strong claims about limitations without considering standard constructions

3. Restrictive Definition of Disentanglement:
- Their requirement that group actions affect single dimensions only is mathematically naive:
  * Ignores fundamental topological obstructions
  * Cannot represent natural symmetries like $SO(3)$ which require coupling between dimensions
  * Confuses independence of factors with independence of dimensions

4. Overcomplicated Solutions:
- Their "conformal mapping" solution is an elaborate way to implement simple circular topology
- The proofs about limitations of existing methods could be greatly simplified if they stated their assumptions clearly
- The mathematical machinery introduced is disproportionate to what's actually being done (mapping to $S^1$)

**Questions:**

1. Why do you require symmetry transformations to act on single dimensions of the latent space? This seems unnecessarily restrictive and rules out many natural symmetries (like 3D rotations) that inherently couple multiple dimensions. Have you considered defining disentanglement in terms of independent factors rather than independent dimensions?

2. In Proposition 2.1 and surrounding theorems, your treatment of matrix groups and exponential maps is unclear. Could you clarify:
   - What exactly is $GL'(n)$ and how does it relate to the Lie algebra $\mathfrak{gl}(n)$?
   - Why do you apply exponential maps to group elements rather than Lie algebra elements?
   - How do you justify the equation $e^g z = eIgz + v'$?

3. Your experiments with 3D shapes use very restricted rotations (15 discrete values). Have you considered testing your method on datasets with full SO(3) rotations? This would seem important given that many real-world applications involve unrestricted 3D transformations.

4. Your framework seems to work well for cyclic, independent transformations, but many real-world symmetries don't decompose this way. Have you explored how your method might be extended to handle more general group actions, or is the restriction to independent cyclic groups fundamental to your approach?

---

> ### Author Response · Authors · 2024-11-25
> **Common Comment for Reviewer QFQU**
>
> To help reviewers clearly understand our motivation and goal, we provided the following common comments:
> 1. Motivation: An explanation of why we focus on the restricted condition of a single dimension.
> 2. Definition: A precise mathematical definition of consistent symmetry.
> ### Motivation: Dimension-wise disentangled representation
> > Disentanglement learning based on VAE has focused on **a single dimension** to contain the information of a single factor [5-11]. Group-theory-based works [12-16] have pursued the same objective, and evaluation metrics [3-6] for disentangled representations have also been developed to assess **how consistently a single factor can be represented in a single dimension**. Representing a single factor within a single dimension has notable strengths in terms of **interpretability** [1]. However, even with very simple data (sprites, 3D Shapes, etc.), this objective has yet to be fully achieved [5-16]. Thus, before addressing real-world data and general cases, **it is necessary to first discuss how to inject inductive bias in simple cases effectively**. Therefore, our work focuses on these highly restricted scenarios.
> ### Definition: Consistent Symmetry and Inconsistent Symmetry
> > Let us assume, $G_F = \\{e, g_{F_j}, g_{F_j}^2, \\ldots, g_{F_j}^k \\}$, $F_j = \\{ F_j^0, F_j^1, \\ldots, F_j^k \\}$, subset $F_j^\prime \subset  F_j$, and a function $q_\phi \circ \Omega: F \rightarrow Z$,. Then group $G_{F_j}$ acted on $F_j^\prime$ as follows: $F_j^{i+1} = g_{F_j} \circ F_j^i$. Through composite function $q_\phi \circ \Omega$, the equation $q_\phi \circ \Omega (F_j^{i+1}) = q_\phi \circ \Omega (g_{F_j} \circ F_j^i)$ translate to $z_j^{i+1} = g_{z_j}^{(i) \\rightarrow (i+1)} \circ z_j^i$. Then we define the **consistent symmetry** as $g_{z_j}^{(i) \\rightarrow (i+1)}$ are **identical regardless of $i$**, otherwise  $g_{z_j}^{(i) \\rightarrow (i+1)}$is referred to as an inconsistent symmetry.
> >
> > We revised it in ‘Definition 2.1’ in the revised version.
> ---
>
> > ### References
> >
> > [1] Yoshua Bengio. Representation learning: a review and new perspectives. TPAMI, 2013.
> >
> > [2] Xin Wnag. Disentangled Representation Learning, TPAMI 2022.
> >
> > [3] Abhishek Kumar. VARIATIONAL INFERENCEOF DISENTANGLED LATENT CONCEPTS FROM UNLABELED OBSERVATIONS. ICLR, 2018.
> >
> > [4] Cian Eastwood and Christopher K. I. Williams. A framework for the quantitative evaluation of disentangled representations. In International Conference on Learning Representations, 2018.
> >
> > [5] Irina Higgin,. beta-vae: Learning basic visual concepts with a constrained variational framework. In ICLR, 2017.
> >
> > [6] Hyunjik Kim and Andriy Mnih. Disentangling by factorising., ICML, 2018.
> >
> > [7] Diederik P Kingma and Max Welling. Auto-encoding variational bayes, 2013.
> >
> > [8] Ricky T. Q., Isolating sources of disentanglement in variational autoencoders., NeurIPS, 2018.
> >
> > [9] Huajie Shao, ControlVAE: Controllable variational autoencoder., ICML, 2020.
> >
> > [10] Yeonwoo Jeong and Hyun Oh Song. Learning discrete and continuous factors of data via alternating disentanglement., ICML, 2019.
> >
> > [11] Huajie Shao, Rethinking controllable variational autoencoders., CVPR, June 2022.
> >
> > [12] Xinqi Zhu, Commutative lie group VAE for disentanglement learning. CoRR, abs/2106.03375, 2021.
> >
> > [13] Loek Tonnaer, Quantifying and learning linear symmetry-based disentanglement., ICML, 2022.
> >
> > [14] Hee-Jun Jung, CFASL: Composite factor-aligned symmetry learning for disentanglement in variational autoencoder. TMLR, 2024.
> >
> > [15] Nikita Balabin, Disentanglement learning via topology., ICML, 2024.
> >
> > [16] Tao Yang, Groupifyvae: from group-based definition to vae-based unsupervised representation disentanglement. CoRR, abs/2102.10303, 2021.
> >
> > [17] Symmetry-based disentangled representation learning requires interaction with environments. NeurIPS 2019
> >
> > [18] Homomorphism Autoencoder — Learning Group Structured Representations from Observed Transitions. ICML 2023.
> >
> > [19] Neural Fourier Transform: A General Approach to Equivariant Representation Learning. ICLR 2024.
> >
> > [20] Latent Space Symmetry Discovery. ICML 2024.
> >
> > [21] Geonho Hwang, MAGANet: Achieving combinatorial Generalization by modeling a group action., ICML, 2023.
> >
> > [22] https://github.com/yvan/cLPR
> >
> > [23] Jaehoon Cha and Jeyan Thiyagalingam. Orthogonality-enforced latent space in autoencoders: An approach to learning
> >
> > [24] Vankov, I. I. and Bowers, J. S. Training neural networks to encode symbols enables combinatorial generalization. Philosophical Transactions of the Royal Society B, 375 (1791):20190309, 2020.
> >
> > [25] Irina Higgins, Towards a definition of disentangled representations. CoRR, abs/1812.02230, 2018.
> >
> > [26] Yingheng Wang, InfoDiffusion: Representation Learning Using Information Maximizing Diffusion Models, ICMLR, 2023.
> >
> > [27] Tero Karras, A Style-Based Generator Architecture for Generative Adversarial Networks, CVPR, 2019.

---

> ### Author Response · Authors · 2024-11-25
> **Response to Reviwer QFQU # 1**
>
> R2 W1.
> > The mathematical treatment of Lie groups and their actions is confused, particularly in their handling of $GL^\prime(n)$ and matrix exponentials
>
> R2 A1.
> > We apologize for the misrepresentation that caused misunderstanding. We define the $GL^\prime(n)$ as $GL^\prime(n) = \\{ e^M | M \in \mathbb{R}^{n\times n}\\}$ (we highlight it on L146 in the revised version).
>
> ---
>
> R2 W2.
> > The fundamental assumption that symmetries must act on single dimensions is too restrictive and ignores basic topological constraints (e.g., impossibility of faithfully representing SO(3), i.e. symmetries of 3D objects, through independent SO(2) actions, i.e. the largest compact group that can act on a single latent dim.)
>
> R2 A2.
> > As mentioned in the ‘Common Comments for Reviewer QFQU’ to avoid repetition. In sum, we focus on dimension-wise disentangled representation, which is a setting studied in various research for disentangled representation.
> Additionally, we **extended the experiment** to cLPR [22] dataset, which consists of rotations of a cube **(symmetries of 3D objects)**. Then we present the **qualitative results on page 27** of the revised version. Our methods (CMCS-GT and CMCS-super) **outperform** Homomorphism VAE [18], which directly uses the SO(3) during training over the cyclic group.
>
> ---
>
>
> R2. W3.
> > The "conformal mapping" solution is unnecessarily complicated for what is essentially just mapping to circular latent spaces
>
> R2. A3.
> > The **essential point** of mapping to circular latent spaces is to **achieve a cyclic structure in a single dimension** rather than in the multi dimensions. As mentioned in **‘Common Comments for Reviewer QFQU’**, we are considering the highly restricted case where $z=\mathbb{R}$. It means that our goal is to identify an arbitrary group $G$ that acts on $z=\mathbb{R}$ such that it **satisfies $g_z \circ \infty = -\infty$**. To implement this arbitrary group $G$, we map a single dimension to circular latent spaces through a bijective function for an **isomorphism** to preserve the symmetry structure of a single dimension ($f(g⋅z)=so(2)⋅f(z)$.)
> >
> > In the case you mentioned, where $Z= \mathbb{R}^2$ and $G=SO(2)$, applying the group action results in the alteration of values in **two dimensions**, which falls outside the scope of our approach (dimension-wise disentangled representations).
>
> ---
> R2. W4.
> > The method only handles independent cyclic symmetries, but this crucial limitation isn't clearly acknowledged. Dataset choice carefully avoids challenging cases (like full 3D rotations) that would reveal the method's limitations
>
> R2 A4.
> > We want to note that **we do not intentionally exclude** full 3D rotation, following more coverage of usual benchmarks of dimension-wise disentanglement learning.
> >
> > Additionally, to provide intuition to the reviewer's hypothetical question, we extended the experiment to cLPR [22] dataset, which consists of rotations of a cube **(symmetries of 3D objects)**. Then we present the **qualitative results on page 27** of the revised version. Our methods (CMCS-GT and CMCS-super) **outperform** Homomorphism VAE [18], which directly uses the SO(3) during training over the cyclic group.
>
> ---
>
> R2 W5.
>
> > Presents as a general solution for symmetries in latent spaces when it's actually quite restricted
>
> R2 A5.
>
> > Our main consideration is the **dimension-wise disentangled representations** as [5- 16]. Our research is studied for the restricted, but required setting studied in [5-16] for disentanglement learning.
>
> ---
>
> R2 W6.
> >* Insufficient discussion of prior work on group actions in latent spaces
> >* Doesn't clearly position their contribution relative to existing symmetry-based approaches
> >* The core idea of using group actions for consistent representations isn't novel (Caselles-Dupré et al "[17] Symmetry-based disentangled representation learning requires interaction with environments." NeurIPS 2019), though their specific implementation for cyclic groups is
>
> R2 A6.
> > **Two types** of symmetry-based prior works for disentangled representation exist: **dimension-wise** and vector-wise disentangled representations [18–20]. Dimension-wise methods [12–17], which are our main focus, utilize symmetries to inject inductive bias. However, there is **insufficient discussion** about which symmetry representations are **appropriate**. Furthermore, **the work mentioned in [17] falls under ‘Case 3’** of our paper, leading to a loss of symmetry information (Corollary 2.5). In contrast, we propose guidelines **for selecting appropriate symmetry representations for dimension-wise disentangled representations.**

---

> ### Author Response · Authors · 2024-11-25
> **Response to Reviewer QFQU # 2**
>
> R2 W8.
> > The proofs about limitations of existing methods are flawed due to overly restrictive assumptions
>
> R2 A8.
> > We agree that our work is based on restrictive assumptions, **but we focus on dimension-wise disentangled representation** [5–16]. Additionally, the major evaluation metrics [3–6] for disentangled representation are based on dimension-wise methods [2]. The evaluation metrics we use estimate the consistency of how a single dimension of the latent vector encapsulates a single factor [3–6].
> >
> > Also, **we summarize our justification of conditions** as follows: our common conditions of three cases are 1) There exists an equivariant function $q_\phi \circ \Omega: \mathcal{F} \rightarrow \mathcal{Z}$ mapping **fully disentangled** factor and latent space. 2) $\mathcal{Z}$ is a $G_z$-set that is a symmetry group action on $\mathcal{Z}$. 3) Group element $g_z$ only affects to a single dimension value of latent vector $z$, where $g_z \in G_z$ **for dimension-wise disentangled representation**.
> >
> > For case 1, the conditions are 4) the symmetry group $G_z(GL^\prime(n))$ acting on latent vector space is defined as a subgroup of the General Linear group, implemented with matrix exponential. 5) For $g^k \in \mathbb{R}^{n \times n}$ and $g=\prod_k g^k$, $g^k$ only affects the $k^{th}$ dimension value of vector $z$. For case 2, the assumption is 6) $g_z$ is a vector.
> >
> > We left these are in the appendix A.
>
> ---
>
> R2 W9.
> > The "consistent symmetry" definition could be much more simply stated
>
> R2 A9.
>
> > We left the **consistent symmetry** definition mathematically in **"Common Comment for All Reviewers"** to avoid repetition.
>
> ---
>
> R2 W10.
> >* Results are only shown for cases that fit their restricted framework
> >* Missing experiments with more challenging symmetries (e.g. more general 3D rotations).
> >* No comparison with methods specifically designed for handling group actions
>
> R2 A10.
>
> > Additionally, we extended the experiment to cLPR [22] dataset, which consists of rotations of a cube **(symmetries of 3D objects)**. Then we present the qualitative results on **page 27** of the revised version. Our methods (CMCS-GT and CMCS-super) **outperform** Homomorphism VAE [18], which directly uses the SO(3) during training over the cyclic group.
> > We add the ‘Homomorphism VAE [18] and Groupified VAE [16], which utilizes the same function to implement unit circle as in [17], methods for reviewer’s requirements and **evaluate the beta-VAE, FVM, MIG, and DCI metrics** on the dSprites dataset using ten seeds as shown in Table 6 of the revised version. (page 23).
> >
> >\\begin{array}{c|c|c|c|c}
> \\hline
> dSprites & beta-VAE & FVM & MIG & DCI \\\\
> \\hline
> Homomorphism VAE [18] & 18.80(\\pm 5.75) & 30.24(\\pm 12.18) & 0.39 (\\pm 0.76) & 1.35 (\\pm 1.12) \\\\
> Groupified VAE [16] & 79.30(\\pm 9.23) & 69.75(\\pm 13.66) & 21.03(\\pm 9.20) & 31.08(\\pm 10.87) \\\\
> CMCS-SP & \\textbf{91.40}(\\pm 4.99) & \\textbf{93.74}(\\pm 1.82) & \\textbf{51.02}(\\pm 2.42) & \\textbf{64.69} (\\pm 1.55) \\\\
> \\hline
> \\end{array}
> >
> > As expected, our model is **much better** than Homomorphism VAE in quantitative results (Table 6 on page 23 of the revised version), and also our model contains each axis rotation in a single dimension (Fig. 17 on page 27 of the revised version).
>
> ---
>
>
>
> R2. W11.
> >* Proposition 2.1 and related theorems contain serious mathematical confusions:
> >   1. Write nonsensical equations like $e^gz=eIgz+v^\prime$ where exponentials are applied inconsistently
> >   2. Mix up elements of Lie groups with elements of Lie algebras
> >   3. Attempt to exponentiate group elements rather than Lie algebra elements
> >   4. Fail to properly distinguish between matrix exponential and scalar exponential
>
> R2 A11.
> > We would like to say, that we did not fail to distinguish between matrix exponential and scalar exponential, and we left about his comment. Also, we left the how **equation $e^gz=eIgz+v^\prime$ satisfies**.
> > **The group element $g$ considered in Line 147 is not all the case of $GL(n)$**, but, as mentioned in Proposition 2.1, it is an element of $GL(n)$ that **affects only a single dimension of the latent vector $z$**. While it is trivial that Line 147 would not satisfy the equation in the absence of all the conditions we proposed, however **under the specified conditions**, Line 147 is satisfied. The **power of $g$ represents the multiple times act on the latent vector $z$**. This is demonstrated through **Eq. (9)** in Appendix A.1 (Lines 791-807). Therefore we do not mix up elements of Lie groups and Lie algebras. Because we consider that $g$ only affects a single dimension, it may appear that we do not distinguish between matrix exponential and scalar exponential. However, we do not confuse the two.

---

> ### Author Response · Authors · 2024-11-25
> **Response to Reviewer QFQU # 3**
>
> R2. W12.
> > * Flawed Analysis of Matrix Groups:
> >   * Their critique of using $GL(n)$ ignores well-known solutions in Lie theory:
> >   * Don't consider proper compact subgroups like $O(n)$ where cyclic behavior is natural.
> >   * Miss the fact that elements of $so(n)$ can easily generate cyclic groups via exponential map
> >   * Make unnecessarily strong claims about limitations without considering standard constructions
>
> R2. A12.
> > Our consideration is **dimension-wise disentangled representation** as [5-16]. Also, $O(n)$, and $SO(n)$ affect multi-dimension of latent vectors, so this method is not appropriate to achieve a dimension-wise disentangled representation. Also, we show **outperform** [17, 18] on ‘R2 A10.’.
>
> ---
>
>
> R2. W13.
> > Restrictive Definition of Disentanglement:
> > * Their requirement that group actions affect single dimensions only is mathematically naive:
> >   * Ignores fundamental topological obstructions
> >   * Cannot represent natural symmetries like SO(3) which require coupling between dimensions
> >   * Confuses independence of factors with independence of dimensions
>
> R2. A13.
> > We left comments that our model **covers SO(3) cases** with additional experiments, and we put **references for dimension-wise disentanglement methods**.
> > Additionally, we extended the experiment to cLPR [22] dataset, which consists of rotations of a cube (symmetries of 3D objects). Then we present the qualitative results on **page 27 of the revised version**. Our methods (CMCS-GT and CMCS-super) **outperform** Homomorphism VAE [18], which directly uses the SO(3) during training over the cyclic group. Also, our model contains each axis rotation information **onto a single dimension** of latent vectors.
> > Our method is based on the **dimension-wise disentangled representation [5-17]**, so we do not confuse the independence of factors with the independence of dimensions.
>
> ---
>
>
>
> R2 W14.
>
> > * Their "conformal mapping" solution is an elaborate way to implement simple circular topology
> > * The proofs about limitations of existing methods could be greatly simplified if they stated their assumptions clearly
> > * The mathematical machinery introduced is disproportionate to what's actually being done (mapping to S1)
>
> R2 A14.
>
> > * Although previous works [16, 17, 23] use functions that map a single dimension to a unit circle, these approaches **fall under ‘Case 3,’ as mentioned in our paper**. In contrast, conformal mapping is a **bijective function** for **homomorphism** that preserves the group structure between two different latent vector spaces as shown in Corollary 2.5.
> > * I revised proposition 2.1, theorem 2.2, and theorem 2.3, which were written in a mix of assumptions and conditions, to make them easier to understand. Also, we highlighted these in the revised version (L145-153).
> > * The **essential point** of mapping to circular latent spaces is to **achieve a cyclic structure in a single dimension** rather than in the multi-dimensions. As mentioned in **‘Common Comments for Reviewer QFQU’**, we are considering the highly restricted case where $z=\mathbb{R}$. It means that our goal is to identify an arbitrary group $G$ that acts on $z=\mathbb{R}$ such that it **satisfies $g_z \circ \infty = -\infty$**. To implement this arbitrary group $G$, we map a single dimension to circular latent spaces through a bijective function for an **isomorphism** to preserve the symmetry structure of a single dimension ($f(g⋅z)=so(2)⋅f(z)$.)
>
> ---
>
> R2 Q1.
> > Why do you require symmetry transformations to act on single dimensions of the latent space? This seems unnecessarily restrictive and rules out many natural symmetries (like 3D rotations) that inherently couple multiple dimensions. Have you considered defining disentanglement in terms of independent factors rather than independent dimensions?
>
> R2 QA1.
> > We mentioned in the ‘Common Comments for Reviewer QFQU’ to avoid repetition.
>
> ---
>
> R2 Q2.
> > In Proposition 2.1 and surrounding theorems, your treatment of matrix groups and exponential maps is unclear. Could you clarify:
>
> R2 Q2-1
> > What exactly is GL′(n) and how does it relate to the Lie algebra gl(n)?
>
> R2 QA 2-1
> > To avoid misleading, we modify the definition of the $GL^\prime(n)$ as a subset of $GL(n)$, which is implemented with matrix exponential (we highlight it on L142-143 in the revised version).
>
> ---
>
> Rw Q2-2
> >* Why do you apply exponential maps to group elements rather than Lie algebra elements?
> >* How do you justify the equation egz=eIgz+v′?
>
> R2 QA 2-2
> > We answer it on R2 A11 to avoid repetition.

---

> ### Author Response · Authors · 2024-11-25
> **Response to Reviewer QFQU # 4**
>
> R2 Q2-3 & Q2-4
> > * Your experiments with 3D shapes use very restricted rotations (15 discrete values). Have you considered testing your method on datasets with full SO(3) rotations? This would seem important given that many real-world applications involve unrestricted 3D transformations.
> > * Your framework seems to work well for cyclic, independent transformations, but many real-world symmetries don't decompose this way. Have you explored how your method might be extended to handle more general group actions, or is the restriction to independent cyclic groups fundamental to your approach?
>
> R2 QA 2-3
> > We extend our experiment (SO(3) dataset) and it is demonstrated in Figure 17 on page 27 in the revised paper.
>
> R2 QA 2-4
> > Even though we did not change any methodology of ours for the SO(3) dataset, our model outperform [18], which directly utilizes the SO(3) group during training.

---

> > ### Comment · Area_Chair_U95m · 2024-11-27
> > **Rebuttal Response**
> >
> > Dear Reviewer,
> > Do you mind letting the authors know if their rebuttal has addressed your concerns and questions? Thanks!
> > -AC

---

> > ### Comment · Reviewer_QFQU · 2024-12-02
> > **Cube experiments show some promise**
> >
> > Thank you for your response. What you are defining as *consistent symmetry* is discrete group with a *single generator*, $g = g_{z_j}^{(i) \rightarrow (i+1)}$ (same for all $i$).  Note that, many finite discrete groups have more than one generator (e.g. the symmetry group of a 3D cube). Also, many discrete groups are not finite (e.g. $\mathbb{Z}$) and cannot be mapped to a cyclic group.
> > A2: Thanks for the cube experiments in Fig 17. They rotations seem to kind of work, but shouldn't the final rotations be $2\pi$? The angles seem quite small. Nevertheless, you method does better than Homomorphism VAE and the results seem promising. One could hypothesize that part of the failure is due to the fact that a disentagled rep of rotations around different axes cannot be found, precisely because SO(3) is non-abelian.
> > A3: I understand that the restriction is intentional. My point is that often symmetries you may try to enforce are groups which don't work with this restriction (anything non-compact), or are just non-Abelian, where your assumption of distentanglement doesn't hold. You would first need to argue or prove that the group you are considering is not this way, or prove it won't be a problem.
> >
> > I do think the theorems need to be simplified and the group definitions etc needs to be corrected. That being said, since the method shows partial success even for non-abelian groups I think it is noteworthy and I lean toward accepting it. So I increase my score.

---

### Official Review · Reviewer_D5sa · 2024-11-05

**Soundness:** 2
**Presentation:** 1
**Contribution:** 3
**Rating:** 3
**Confidence:** 2

**Summary:**

This paper analyzes disentanglement learning and combinatorial generalization from the perspective of designing equivariant latent spaces. They define a desirable notion of “consistent symmetry”, which autoencoders trained to equivariantly encode data arising from “latent factors of variation” should satisfy, but show that several choices of latent space cannot satisfy this property. Instead, they propose a method, CMCS, which more carefully designs the latent space (on which the latent factors — modeled as a direct product of cyclic groups — are encouraged to act via addition by the loss function). Overall, it consists of mapping real-valued scalar latents to complex numbers, some kind of grid selection, a choice of action of the cycle group (the step size), and a loss function that encourages the group action on the latent space to be respected. Their method works well on experiments testing disentanglement learning and combinatorial generalization.

**Strengths:**

The authors show that previous approaches to disentangled representations lack the consistent symmetry property, so their results are relevant to existing literature. Moreover, they suggest a novel alternative, which outperforms previous work on their experiments in both the reconstruction loss and disentanglement learning. At a very high level, these seem like valuable and original contributions. However, as detailed below, the presentation of the work needs such improvement that I can’t really understand exactly what the consistent symmetry property even is, how trivial or non-trivial the impossibility results are, or what their method is doing. I hope the rebuttal will clarify these points.

**Weaknesses:**

1. The presentation of the paper is unfortunately very poor. For example, key concepts such as “combinatorial generalization” and “latent factors of variation” are used without explanation, or motivation. Related works are cited, but not explained or summarized, in the introduction. Figure 2 needs a more detailed caption. The presentation of concepts is not mathematically precise, which makes them nearly impossible to understand (see questions). For example, even the definition of Consistent Summary before Section 2.2 is difficult to parse. For this reason, it is possible I have not understand all of the contributions of the paper fully, and I hope the authors’ response will clarify. At the least, the paper would need to be very substantially rewritten (in my opinion) for it to be clear and precise enough for publication.
2. Since disentanglement and latent factors of variation are not motivated or defined, it is difficult to assess the significance of this work, or the novelty of the proposed method.
3. Other writing notes: the datasets should be explained briefly in the main body, not relegated to the appendix, to orient the reader before showing results — especially for a 10 page submission. This is important for evaluating the results. It would also be good to define “conformal” (unless I missed it somewhere).

**Questions:**

1. I don’t understand Figure 2. Are the red arrows supposed to indicate inequality? Why are they different shades of red? The only difference between the left, inconsistent case and the right, consistent case are the arrow colors and the labels on the group elements, but I don’t understand the significance of these differences.
2. I don’t understand the definition of Consistent Symmetry. Is it a specific group element? A property of a model? Mathematically, what does it mean? Intuitively (eg with a concrete example), what does it mean? Perhaps it would also be helpful to formally define “representing the cyclic semantics of a dataset with consistent symmetries”. What is the “pair of latent factors”?
3. Is it possible to define consistent symmetry in terms of the equivariance of $\Lambda$, by choosing the right group (perhaps $G_F \times G_z$ and group action?
 4. How realistic is the assumption that the group acting on the latent factors of variation is the direct product of cyclic groups, especially of known dimensionality and number?
5. One could imagine a latent factor of variation changing with respect to permutations, $S_n$. Assuming it doesn’t cover 100% of cases, would either the impossibility results or the improved method by extensible to a direct product of different groups?
6. At a high level, what is it about the proposed method that circumvents the impossibility result of Theorem A.9 (which also uses the action of addition)?
7. What is the proof technique behind the impossibility results? Is there intuition that can be put in the main body of the work?
8. What exactly is the fixed codebook and fixed grid doing? A figure might be helpful.
9. What does conformal mean in this context? Why is it a useful property for the mapping to have?

---

> ### Author Response · Authors · 2024-11-25
> **Common Comments for Reviewer D5sa**
>
> To help reviewers clearly understand our motivation and goal, we provided the following common comments:
> 1. Motivation: An explanation of why we focus on the restricted condition of a single dimension.
> 2. Definition: A precise mathematical definition of consistent symmetry.
> ### Motivation: Dimension-wise disentangled representation
> > Disentanglement learning based on VAE has focused on **a single dimension** to contain the information of a single factor [5-11]. Group-theory-based works [12-16] have pursued the same objective, and evaluation metrics [3-6] for disentangled representations have also been developed to assess **how consistently a single factor can be represented in a single dimension**. Representing a single factor within a single dimension has notable strengths in terms of **interpretability** [1]. However, even with very simple data (sprites, 3D Shapes, etc.), this objective has yet to be fully achieved [5-16]. Thus, before addressing real-world data and general cases, **it is necessary to first discuss how to inject inductive bias in simple cases effectively**. Therefore, our work focuses on these highly restricted scenarios.
> ### Definition: Consistent Symmetry and Inconsistent Symmetry
> > Let us assume, $G_F = \\{e, g_{F_j}, g_{F_j}^2, \\ldots, g_{F_j}^k \\}$, $F_j = \\{ F_j^0, F_j^1, \\ldots, F_j^k \\}$, subset $F_j^\prime \subset  F_j$, and a function $q_\phi \circ \Omega: F \rightarrow Z$,. Then group $G_{F_j}$ acted on $F_j^\prime$ as follows: $F_j^{i+1} = g_{F_j} \circ F_j^i$. Through composite function $q_\phi \circ \Omega$, the equation $q_\phi \circ \Omega (F_j^{i+1}) = q_\phi \circ \Omega (g_{F_j} \circ F_j^i)$ translate to $z_j^{i+1} = g_{z_j}^{(i) \\rightarrow (i+1)} \circ z_j^i$. Then we define the **consistent symmetry** as $g_{z_j}^{(i) \\rightarrow (i+1)}$ are **identical regardless of $i$**, otherwise  $g_{z_j}^{(i) \\rightarrow (i+1)}$is referred to as an inconsistent symmetry.
> >
> > We revised it in ‘Definition 2.1’ in the revised version.
> ---
>
> > ### References
> >
> > [1] Yoshua Bengio. Representation learning: a review and new perspectives. TPAMI, 2013.
> >
> > [2] Xin Wnag. Disentangled Representation Learning, TPAMI 2022.
> >
> > [3] Abhishek Kumar. VARIATIONAL INFERENCEOF DISENTANGLED LATENT CONCEPTS FROM UNLABELED OBSERVATIONS. ICLR, 2018.
> >
> > [4] Cian Eastwood and Christopher K. I. Williams. A framework for the quantitative evaluation of disentangled representations. In International Conference on Learning Representations, 2018.
> >
> > [5] Irina Higgin,. beta-vae: Learning basic visual concepts with a constrained variational framework. In ICLR, 2017.
> >
> > [6] Hyunjik Kim and Andriy Mnih. Disentangling by factorising., ICML, 2018.
> >
> > [7] Diederik P Kingma and Max Welling. Auto-encoding variational bayes, 2013.
> >
> > [8] Ricky T. Q., Isolating sources of disentanglement in variational autoencoders., NeurIPS, 2018.
> >
> > [9] Huajie Shao, ControlVAE: Controllable variational autoencoder., ICML, 2020.
> >
> > [10] Yeonwoo Jeong and Hyun Oh Song. Learning discrete and continuous factors of data via alternating disentanglement., ICML, 2019.
> >
> > [11] Huajie Shao, Rethinking controllable variational autoencoders., CVPR, June 2022.
> >
> > [12] Xinqi Zhu, Commutative lie group VAE for disentanglement learning. CoRR, abs/2106.03375, 2021.
> >
> > [13] Loek Tonnaer, Quantifying and learning linear symmetry-based disentanglement., ICML, 2022.
> >
> > [14] Hee-Jun Jung, CFASL: Composite factor-aligned symmetry learning for disentanglement in variational autoencoder. TMLR, 2024.
> >
> > [15] Nikita Balabin, Disentanglement learning via topology., ICML, 2024.
> >
> > [16] Tao Yang, Groupifyvae: from group-based definition to vae-based unsupervised representation disentanglement. CoRR, abs/2102.10303, 2021.
> >
> > [17] Symmetry-based disentangled representation learning requires interaction with environments. NeurIPS 2019
> >
> > [18] Homomorphism Autoencoder — Learning Group Structured Representations from Observed Transitions. ICML 2023.
> >
> > [19] Neural Fourier Transform: A General Approach to Equivariant Representation Learning. ICLR 2024.
> >
> > [20] Latent Space Symmetry Discovery. ICML 2024.
> >
> > [21] Geonho Hwang, MAGANet: Achieving combinatorial Generalization by modeling a group action., ICML, 2023.
> >
> > [22] https://github.com/yvan/cLPR
> >
> > [23] Jaehoon Cha and Jeyan Thiyagalingam. Orthogonality-enforced latent space in autoencoders: An approach to learning
> >
> > [24] Vankov, I. I. and Bowers, J. S. Training neural networks to encode symbols enables combinatorial generalization. Philosophical Transactions of the Royal Society B, 375 (1791):20190309, 2020.
> >
> > [25] Irina Higgins, Towards a definition of disentangled representations. CoRR, abs/1812.02230, 2018.
> >
> > [26] Yingheng Wang, InfoDiffusion: Representation Learning Using Information Maximizing Diffusion Models, ICMLR, 2023.
> >
> > [27] Tero Karras, A Style-Based Generator Architecture for Generative Adversarial Networks, CVPR, 2019.

---

> ### Author Response · Authors · 2024-11-25
> **Response to Reviewer D5sa #1**
>
> R1 W1.
> > For example, key concepts such as “combinatorial generalization” and “latent factors of variation” are used without explanation, or motivation.
>
> R1 A1.
> > **Latent factors of variation** is a very common expression in the generative models [26, 27], which are components of the composition of objects, such as shape, x-position, and y-position of datasets.
> > **Combinatorial generalization**, the capacity to comprehend and create new combinations of familiar elements, is regarded as a fundamental capability of the human mind and a significant challenge for neural network models [21, 24].
>
> ---
>
> R1. W2.
> > Related works are cited, but not explained or summarized, in the introduction.
>
> R1 A2.
>  > We have addressed most of the relevant references in the introduction, particularly those related to disentanglement learning and combinatorial generalization. Unfortunately, due to space constraints, we are unable to include additional references for a broader audience.
>
> ---
>
> R1 W3.
> > Figure 2 needs a more detailed caption. The presentation of concepts is not mathematically precise, which makes them nearly impossible to understand (see questions). For example, even the definition of Consistent Summary before Section 2.2 is difficult to parse. For this reason, it is possible I have not understand all of the contributions of the paper fully, and I hope the authors’ response will clarify.
>
> R1 A3.
> > We define **consistent symmetry** mathematically, please check the **‘Common Comments for Reviewer D5sa’**, also, **we revised** Figure 2 in the revised version.
>
> ---
>
> R1 W4.
> > Since disentanglement and latent factors of variation are not motivated or defined, it is difficult to assess the significance of this work, or the novelty of the proposed method.
>
> R1 A4.
> > We left the comment of disentanglement and latent factors in the R1 A1., and motivation is in the ‘Common Comments for Reviewer D5sa’.
>
> ---
>
> R1 W5.
> > Other writing notes: the datasets should be explained briefly in the main body, not relegated to the appendix, to orient the reader before showing results — especially for a 10 page submission. This is important for evaluating the results.
>
> R1 A5.
> > The dSprites, 3D shapes, and MPI3d datasets we used are very common datasets for disentanglement learning. Even though the 10 page submission, our contributions are not easy to reduce for the main paper. So we left the datasets information in the Appendix.
>
> ---
>
> R1. Q1.
> > I don’t understand Figure 2. Are the red arrows supposed to indicate inequality? Why are they different shades of red? The only difference between the left, inconsistent case and the right, consistent case are the arrow colors and the labels on the group elements, but I don’t understand the significance of these differences.
>
> R1. QA1.
> > As we define the **inconsistent symmetry on the ‘Common Comments for Reviewer D5sa’**, the inconsistent symmetries depend on the objects, **so the different color arrows refer to inconsistent symmetry in Fig. 2.** We revise the Fig. 2. in the revised paper.
>
> ---
>
> R1. Q2.
> > I don’t understand the definition of Consistent Symmetry. Is it a specific group element? A property of a model? Mathematically, what does it mean? Intuitively (eg with a concrete example), what does it mean? Perhaps it would also be helpful to formally define “representing the cyclic semantics of a dataset with consistent symmetries”. What is the “pair of latent factors”?
>
> R1. QA2.
> > We define the **consistent and inconsistent symmetry** mathematically in **‘Common Comments for Reviewer D5sa’.**
>
> ---
>
> R1. Q3.
> > Is it possible to define consistent symmetry in terms of the equivariance of Λ, by choosing the right group (perhaps GF×Gz and group action?
> R1 QA3.
> > Yes, as we defined in ‘Common Comments for Reviewers D5sa’ the consistent symmetry is eventually defined by a equivariant function.
>
> ---
>
> R1. Q4.
> >How realistic is the assumption that the group acting on the latent factors of variation is the direct product of cyclic groups, especially of known dimensionality and number?
>
> R1. QA4.
> >Many works use the SO(2) for disentangled representation, but these are still limited in fine-grained semantic composed dataset. But that is the challenge of disentanglement and combinatorial generalization to extend task from simple datasets to real scenes.
>
> ---
>
> R1. Q5.
> > One could imagine a latent factor of variation changing with respect to permutations, Sn. Assuming it doesn’t cover 100% of cases, would either the impossibility results or the improved method by extensible to a direct product of different groups?
>
> R1. QA5.
> > In our opinion, if the dataset composes a composition of specific groups then the model could represent the symmetries. Also, our model decomposes each symmetry into a single dimension, **so our method is likely to decompose different groups as shown in our extended experiment with the cLPR dataset.**

---

> ### Author Response · Authors · 2024-11-25
> **Response to Reviewer D5sa #2**
>
> R1. Q6.
> >At a high level, what is it about the proposed method that circumvents the impossibility result of Theorem A.9 (which also uses the action of addition)?
>
> R1, QA6.
> >As shown in Theorem A.9, the vector addition can not represent the cyclic structure over a single dimension. As mentioned in **‘Common Comment for all reviewers’, our goal is to identify an arbitrary group $G$ that acts on $z=\mathbb{R}$ such that it resembling $\infty \rightarrow -\infty$.  As you pointed out, in the case where $Z = \mathbb{R}$, it is not possible to identify a cyclic group $G$ with addition. However, the key aspect of our method lies in finding g that satisfies $f(g⋅z)=so(2)⋅f(z)$ using conformal mapping. While we do not know an arbitrary $G$, we have implemented a method that satisfies the above condition.
> > The **essential point** of mapping to circular latent spaces is to **achieve a cyclic structure in a single dimension** rather than in the multi dimensions. As mentioned in **‘Common Comments for Reviewer D5sa’**, we are considering the highly restricted case where $z=\mathbb{R}$. It means that our goal is to identify an arbitrary group $G$ that acts on $z=\mathbb{R}$ such that it **satisfies $g_z \circ \infty = -\infty$**. To implement this arbitrary group $G$, we map a single dimension to circular latent spaces through a bijective function for an **isomorphism** to preserve the symmetry structure of a single dimension ($f(g⋅z)=so(2)⋅f(z)$.)
>
>
> ---
>
> R1. Q7.
> >What is the proof technique behind the impossibility results? Is there intuition that can be put in the main body of the work?
>
> R1 QA7.
> >Our intuition is started from a dimension-wise disentanglement representation as we mentioned in Line 134-135.
> > So our common conditions of three cases are 1) There exists an equivariant function $q_\phi \circ \Omega: \mathcal{F} \rightarrow \mathcal{Z}$ mapping fully disentangled factor and latent space. 2) $\mathcal{Z}$ is a $G_z$-set that is a symmetry group action on $\\mathcal{Z}$. 3) Group element $g_z$ only affects to a single dimension value of latent vector $z$, where $g_z \in G_z$.
> >
> > For case 1, the conditions are 4) the symmetry group $G_z(GL^\prime(n))$ acting on latent vector space is defined as a subgroup of the General Linear group, implemented with matrix exponential. 5) For $g^k \in \mathbb{R}^{n \times n}$ and $g=\prod_k g^k$, $g^k$ only affects the $k^{th}$ dimension value of vector $z$. For case 2, the assumption is 6) $g_z$ is a vector.
> For case 2, the assumption is 6) $g_z$ is a vector.
> These intuition and conditions are not enough space to pu in the main body of the work, so we left these are in the appendix.
>
> ---
>
> R1. Q8.
> > What exactly is the fixed codebook and fixed grid doing? A figure might be helpful.
>
> R1. QA8.
> > First, the fixed codebook and fixed grid are the same components to **enforce the angle space**. The fixed grid enforces the equations (5), and (6) through a **n^{th} group unity** as we defined in ‘Group action and $G^c$-set’ paragraph (L209-212). As the n^{th} group unity is finite and discrete, so we also **enforce the angle space to be discrete through the fixed grid (fixed codebook)**.
>
> ---
>
> R1. Q9.
> >What does conformal mean in this context? Why is it a useful property for the mapping to have?
>
> R1. QA9.
> >‘Conformal (or angle-preserving) mapping’ is a mathematically defined term as: the function locally preserves angles. We utilize this function for 1) Cayley transform (conformal mapping) maps the $\mathbb{R} \rightarrow \mathbb{S}^1$, for applying $n^{th}$ root unity **for cyclic structure**, 2) conformal mapping is **a bijective** then it avoids the problem of ‘case 3’ (L165-175).

---

> ### Comment · Area_Chair_U95m · 2024-11-27
> **Rebuttal Response**
>
> Dear Reviewer,
> Do you mind letting the authors know if their rebuttal has addressed your concerns and questions? Thanks!
> -AC

---

> > ### Comment · Reviewer_D5sa · 2024-11-27
> > **Thanks for the response**
> >
> > Thanks to the reviewers for their point-by-point response. I appreciate the clarifications added to the submission and better understand the setting now, but still feel that the paper’s presentation needs significant work. Even the new definition of consistent symmetry is still not very rigorous, and such issues pervade the submission. For example, a complete definition should specify the spaces that each object lives in (e.g. $g_{F_j} \in G$, $F_j^I \in ?$), and should also define $\Omega$ and $q_\phi$ separately. Figure 2 could also still use work, eg demonstrating *why* the consistent symmetry is more desirable than the inconsistent symmetry (since the effect on the box with the square in it is the same). The authors also claim that they cannot provide a more general introduction, or short descriptions of the evaluated datasets, due to the 10 page limit — but 10 pages is already long for an ML conference, and other papers manage to find space to perform these critical tasks. Providing a paragraph or two of background that *summarizes* the most relevant related work for a slightly more general audience, rather than a long list of unexplained citations recognized only by domain experts, is a crucial component of writing a good ML paper. For example, I randomly picked reference [13] “Quantifying and learning linear symmetry-based disentanglement”, and found their first three paragraphs provided a much more accessible background, emphasizing pedagogy rather than quantity of citations. [13] also briefly describes their datasets.
> >
> > Outside of the presentation, the assumption (noted by other reviewers as well as the authors themselves) that the group need act on one-dimensional latent spaces seems quite restricted, and it is not surprising that many groups cannot act consistently on one-dimensional spaces. I acknowledge that the experimental results (to a non-expert) seem ok, and that it is a useful observation to make — but attaining sufficient novelty for publication would then have to come from the solution presented, and using a conformal mapping seems like overkill when one can just go to a slightly higher dimensional latent space. Overall, I will retain my rating, but also the low confidence level because I am still not very familiar with the background literature.

---

> > > ### Author Response · Authors · 2024-11-28
> > > **Response to Reviewer D5sa #3**
> > >
> > > Reviewer’s Comment 1
> > > > For example, a complete definition should specify the spaces that each object lives in (e.g. gFj∈G, FjI∈?), and should also define Ω and qϕ separately.
> > >
> > > Author’s Answer 1
> > > > We apologize for not referring to the notations. Each notation is defined **in Lines 90-101** in the revised paper.
> > >
> > > ---
> > >
> > > Reviewer’s Comment 2
> > > > Figure 2 could also still use work, eg demonstrating why the consistent symmetry is more desirable than the inconsistent symmetry (since the effect on the box with the square in it is the same).
> > >
> > > Author’s Answer 2
> > > > Intuitively, **equivariant models** offer a significant advantage in learning symmetries [12-20]. From this perspective, achieving consistent symmetries suggests that the model closely approximates an equivariant model. Consequently, consistent symmetry is preferable to inconsistent symmetry when learning symmetries as an inductive bias.
> > >
> > > ---
> > >
> > > Reviewer’s Comment 3
> > > > The authors also claim that they cannot provide a more general introduction, or short descriptions of the evaluated datasets, due to the 10 page limit — but 10 pages is already long for an ML conference, and other papers manage to find space to perform these critical tasks. Providing a paragraph or two of background that summarizes the most relevant related work for a slightly more general audience, rather than a long list of unexplained citations recognized only by domain experts, is a crucial component of writing a good ML paper. For example, I randomly picked reference [13] “Quantifying and learning linear symmetry-based disentanglement”, and found their first three paragraphs provided a much more accessible background, emphasizing pedagogy rather than quantity of citations. [13] also briefly describes their datasets.
> > >
> > > Author’s Answer 3
> > > > We appreciate your suggestion to include backgrounds and datasets in the main paper to better address a general audience. However, our contributions are too extensive to be fully covered in the main paper, as we address **two major tasks**: (1) disentanglement learning and (2) combinatorial generalization. While most studies focus on a single task, our work tackles both, resulting in an experimental section that spans nearly five pages. To accommodate the general audience, we will provide a **summary of the backgrounds and datasets in the Appendix**. To make it improved, we will add a brief goal and distinguished features briefly.
> > >
> > > ---
> > >
> > > Reviewer’s Comment 4
> > > > Outside of the presentation, the assumption (noted by other reviewers as well as the authors themselves) that the group need act on one-dimensional latent spaces seems quite restricted, and it is not surprising that many groups cannot act consistently on one-dimensional spaces. I acknowledge that the experimental results (to a non-expert) seem ok, and that it is a useful observation to make — but attaining sufficient novelty for publication would then have to come from the solution presented, and using a conformal mapping seems like overkill when one can just go to a slightly higher dimensional latent space.
> > >
> > > Author’s Answer 4
> > > > As we mentioned in the **‘Common Comments for Reviewer D5sa’**, our focus is on the dimension-wise disentangled representation as shown in [5-17]. The essential point of conformal mapping is discussed in **"R1 QA6"**. Additionally, the result of controlling multiple dimensions (Homomorphism VAE [18]) performs **worse than our model**, and we have **extended** our qualitative results on cLPR [22], which consists of **SO(3) symmetries**, on page 27 of the revised paper.
> > > >
> > > >\\begin{array}{c|c|c|c|c}
> > > \\hline
> > > dSprites & beta-VAE & FVM & MIG & DCI \\\\
> > > \\hline
> > > Homomorphism VAE [18] & 18.80(\\pm 5.75) & 30.24(\\pm 12.18) & 0.39 (\\pm 0.76) & 1.35 (\\pm 1.12) \\\\
> > > Groupified VAE [16] & 79.30(\\pm 9.23) & 69.75(\\pm 13.66) & 21.03(\\pm 9.20) & 31.08(\\pm 10.87) \\\\
> > > CMCS-SP & \\textbf{91.40}(\\pm 4.99) & \\textbf{93.74}(\\pm 1.82) & \\textbf{51.02}(\\pm 2.42) & \\textbf{64.69} (\\pm 1.55) \\\\
> > > \\hline
> > > \\end{array}

---

### Author Response · Authors · 2024-11-25
**Response to all Reviewers**

Dear, all Reviewers, We appreciate your valuable and fruitful comments to improve this research. The abbreviation ‘R’ refers to a reviewer, ‘A’ refers to an answer, and 'QA’ refers to an answer to the question.

---

### Note · Authors · 2025-01-23

I have read and agree with the venue's withdrawal policy on behalf of myself and my co-authors.